



# Impact of capillary rise and recirculation on crop yields

*Joop Kroes[1], Iwan Supit[1,2] Jos Van Dam[3], Paul Van Walsum[1], Martin Mulder[1]*

[1] *Wageningen University & Research - Environmental Research (Alterra)*

[2] *Wageningen University & Research – Chair Water Systems and Global Change*

[3] *Wageningen University & Research – Chair Soil Physics and Land Management*

*Abstract*

This paper describes impact analyses of various soil water flow regimes on grass, maize and potato yields in the Dutch delta, with a focus on upward soil water flows capillary rise and recirculation towards the rootzone. Flow regimes are characterised by soil composition and groundwater depth and derived from a national soil database. The intermittent occurrence of upward flow and its influence on crop growth are simulated with the combined

SWAP-WOFOST model using various boundary conditions. Case studies and model experiments are used to illustrate impact of upward flow on yield and crop growth. This impact is clearly present in situations with relatively shallow groundwater levels (85% of the Netherlands), where capillary rise is the main flow source; but also in free-draining situations the impact of upward flow is considerable. In the latter case recirculated percolation water is

the flow source. To make this impact explicit we implemented a synthetic modelling option that stops upward flow from reaching the root zone, without inhibiting percolation. Such a hypothetically moisture-stressed situation compared to a natural one in the presence of shallow groundwater shows mean yield reductions for grassland, maize and potatoes of respectively 25, 4 and 15 % or respectively about 3.2, 0.5 and 1.6 ton dry matter per ha.

About half of the withheld water behind these yield effects comes from recirculated percolation water as occurs in free drainage conditions and the other half comes from increased upward capillary rise. Soil water and crop growth modelling should consider both capillary rise from groundwater and recirculation of percolation water as this improves the accuracy of yield simulations. This also improves the accuracy of the simulated groundwater

recharge: neglecting these processes causes overestimates of 17% for grassland and 46% for potatoes, or 70 and 34 mm a$^{-1}$, respectively.





*1. Introduction*


Crop growth strongly depends on soil moisture conditions. Climate variables determine these conditions through rain that penetrates directly into the root zone or comes available via lateral flow. The moisture distribution in the soil strongly depends on soil physical properties that determine vertical flow. Upward soil water flow becomes an especially vital

supply term of a crop when the soil water potential gradient induced by the root-extraction manages to bridge the distance to the capillary fringe, inducing increased capillary rise. In this paper we follow the definition of capillary rise, given by SSA (2008), as the "phenomenon that occurs when small pores which reduce the water potential are in contact with free water". This implies that capillary rise as a source for upward flow to crop roots

requires the presence of a groundwater table. In conditions without a groundwater table there may also be a contribution of upward flow to crop roots through the process of recirculation. Recirculation is a known process discussed already by Feodoroff (Rijtema and Wassink, 1969), but has never been quantified. We quantified recirculation separately from capillary rise using model experiments.

The contribution of (intermittent) upward flow to the total water budget can be significant. For example Kowalik (2006) mentions that during the growing season, for grass the capillary rise induced by root extraction was equal to 90–150 mm for Aquepts Inceptisols and 60–130 mm for Aquepts Histosols. In dry years the induced capillary supply can be 40–50% of the total supply for Histosols, but close to zero for some Inceptisols (Kowalik, 2006).

Babajimopoulos et al. (2007) found that under the specific field conditions about 3.6 mm/day of the water in the root zone originated from the shallow water table, which amounts to about 18% of the water transpired by a maize crop. Fan et al. (2013) analysed the groundwater depth globally and concluded that shallow groundwater influences 22 to 32% of global land area, and that 7 to 17% of this area has a water table within or close to plant

rooting depths, suggesting a widespread influence of groundwater on crops. This is especially the case in delta areas where high population densities occur and agriculture is the predominant land use.

Wu et al. (2015) showed that capillary rise plays a main role in supplying the vegetation throughout the season with water, hence a strong dependence of vegetation upon

groundwater. Han et al. (2015) applied HYDRUS-1D with a simplified crop growth model and concluded for cotton in a north-western part of China that capillary rise from groundwater contributes almost to 23% of crop transpiration when the average groundwater depth is 1.84 m. According to Geerts et al. (2008) the contribution from capillary rise to the quinoa production in the Irpani region (Peru), ranges from 8 to 25% of seasonal crop

evapotranspiration (ETc) of quinoa, depending mostly on groundwater table depth and amount of rainfall during the rainy season. The contribution from a groundwater table




located approximately 1.5 to 2 m deep may represent up to 30% of the soybean water requirements in sandy pampas (Videla Mensegue et al., 2015).

In the Netherlands the average groundwater table is less than 2 meter below the soil surface in 85% of the area (De Vries, 2007), where root extraction can induce capillary rise from groundwater. Wesseling and Feddes (2006) report that in summers with a high evapotranspiration demand, crops partially depend on water supply from soil profile storage and induced capillary rise. Van der Gaast et al. (2009), applying the method of Wesseling

(1991), found for the Netherlands a maximum capillary flow of 2 mm/d to the root zone in loamy soils where the groundwater level is at 2.5 meter below the soil surface.

Although the contribution of capillary rise to the total water budget can be significant, it is an often neglected part of the crop water demand in situations of shallow groundwater levels (Awan et al., 2014). The capillary properties of a soil strongly depend on soil type. Rijtema

(1971) estimated that loamy soils have an almost 2 times higher capillary rise than sandy soils.

Integrated approaches are needed to relate water availability to crop yield prognosis (Van der Ploeg and Teuling, 2013; Norman, 2013). The importance of capillary rise as supplier of

water to crops has been shown by many researchers (e.g. Huo et al., 2012; Talebnejad, and Sepaskhah, 2015; Han et al., 2015); however we found only a few studies that use an integrated modelling approach (Xu et al., 2013; Zipper et al. 2015) to quantify capillary rise for different hydrological conditions (including free drainage) using physically based approaches. In this study we explicitly consider the effect of crop type, soil type, weather

year and drainage condition on capillary rise. Zipper et al. (2015) introduced the concept of groundwater yield subsidy ss the increase in yield in the presence of shallow groundwater compared to free drainage conditions. Following their line we introduce the concept of soil moisture yield subsidy as additional yield increase in free drainage conditions due to recirculation of percolated soil moisture.


The driving force for induced capillary rise is the difference in soil water potential, referred to as heads. There are several models available that solve these head differences in a numerical way. Ahuja et al. (2014) evaluated 11 models commonly applied for agricultural water management. Six of these models use simple 'bucket' approaches for water storage

and have in some cases been extended with more or less empirical options for capillary rise. Five models have the ability to numerically solve Richard's equation for water movement in the soil. Examples are HYDRUS (Šimůnek et al., 2008) and SWAP (Feddes et al., 1988, Van Dam et al., 2008).





We applied the integrated model SWAP-WOFOST to solve head differences and crop yield
simulations. Kroes and Supit (2011) applied the same integrated model to quantify impact of
increased groundwater salinity on drought and oxygen of grassland yields in the
Netherlands. They recommended further analyses using different crops and different
boundary conditions. We now apply this model with different boundary conditions using 45
years of observed weather and three different crops. For the lower boundary we use
different hydrologic conditions that influence the vertical flow. For the soil system itself we
use a wide range of soil physical conditions. The importance of the soil system was already
stated by several authors like Supit (2000). We build on their suggestions and apply the
tools for different crops and boundary conditions. Before we applied the model to different
boundary conditions we validated it at field scale.


This paper quantifies the effects of (intermittent) upward flow on crop growth under different
conditions of soil hydrology, soil type and weather. The effects are separately quantified in
terms of flow source, namely capillary rise and recirculated percolation water. To make this
separate quantification we performed a numerical experiment introducing a synthetic model
option. We studied forage maize, grassland and potatoes and we hypothesize that
neglecting upward flow will result in neglecting a considerable amount of soil moisture that is
available for crop growth. We quantify this amount and show the importance of including
upward flow for crop growth modelling. Our main research questions are: i) Can upward flow
with as source capillary rise and recirculated percolation water be quantified separately?, ii)
What is the contribution of capillary rise and recirculated water to crop yield and
groundwater recharge?

## 2.  Materials and methods


### 2.1 Modelling approach

We applied the coupled SWAP and WOFOST modeling system, using a one day time step.
SWAP (Van Dam et al., 2008; Kroes et al., 2009) is a one-dimensional physically based
transport model for water, heat and solute in the saturated and unsaturated zone, and
includes modules for simulating irrigation practices. The first version of SWAP, called
SWATRE, was developed by Feddes et al. (1978). SWAP simulates the unsaturated and
saturated water flow in the upper part of the soil system, using a numerical solution of the
Richards' equation:

$$\frac{\partial \theta}{\partial t} = \frac{\partial \left[ K(h) \left( \frac{\partial h}{\partial z} + 1 \right) \right]}{\partial z} - S_a(h) - S_d(h) - S_m(h)$$   (1)





where: $\theta$ is volumetric water content (cm$^3$ cm$^{-3}$), $t$ is time (d), $K(h)$ is hydraulic conductivity (cm d$^{-1}$), $h$ is soil water pressure head (cm) and $z$ is the vertical coordinate (cm), taken positively upward, $S_a(h)$ is soil water extraction rate by plant roots (d$^{-1}$), $S_d(h)$ is the extraction rate by drain discharge in the saturated zone (d$^{-1}$) and $S_m(h)$ is the exchange rate with macro pores (d$^{-1}$).


Root water extraction and lateral exchange with surface water are accounted for. In this study we do not use the option to exchange water flow with macro pores.

The soil hydraulics are described by the Mualem–Van Genuchten relations and the potential evapotranspiration is calculated with the Penman–Monteith equation (Allen et al., 1998).

Hydraulic heads supplied by a separate regional hydrological model can be used to simulate interaction between bottom boundary fluxes and groundwater levels. Drainage and infiltration through the lateral boundary account for the flow to surface water. The surface water system is simulated using a simplified, weir controlled, water balance. Note that the surface water system in its turn interacts with the groundwater system. In previous years,

SWAP has been successfully used to study soil-water-atmosphere-plant relationships in many locations with various boundary conditions (e.g. Feddes et al., 1988; Bastiaanssen et al., 2007). See Van Dam et al. (2008) for an overview. A recent list is available at http://www.swap.alterra.nl. Eitzinger et al. (2004), Bonfante et al. (2010), Oster et al. (2012), and Rallo et al. (2012) amongst others tested the model performance.

Van Keulen and Wolf (1986) explained the WOFOST principles and Van Diepen et al. (1989) presented the first WOFOST version. WOFOST is a crop growth simulation model applied in many studies (e.g Rötter, 1993; Van Ittersum et al., 2003; de Wit and Van Diepen, 2008; Supit et al., 2012; De Wit et al., 2012). In WOFOST the crop assimilation is a function of solar radiation and temperature. At 3 times of the day WOFOST calculates radiation

profiles within the canopy using leaf angle distribution and extinction of direct and diffuse light. Next the total assimilation is calculated by integrating the assimilation-light response of single leaves over the canopy and day. The assimilation is reduced when water or nutrient stress occurs. Subsequently, the maintenance respiration is subtracted and the remaining assimilates are partitioned over the plant organs (i.e. leaves, stems, roots and storage

organs). For maize and potatoes the partitioning is development stage dependent. For perennial grass however, a constant partitioning factor is assumed. By integrating the difference between growth and senescence rates over time, dry weights of various plant organs are established.

In SWAP-WOFOST, crop assimilation depends on the ambient $CO_2$ concentration as well

(see: Kroes and Supit, 2011; Supit et al., 2012). To account for unknown residual stress caused by diseases, pests and/or weeds an additional assimilation reduction factor is introduced. The rooting density decreases exponentially with depth. To withdraw water from





deeper soil layers for crop uptake  a form of compensatory root uptake is used in case the upper part of the soil is very dry (Jarvis, 2011). The increasing atmospheric $CO_2$

concentrations during relatively long historical simulation periods (>20years) is accounted for.

*2.2 Case studies for validation*

SWAP-WOFOST is validated using results of 7 case studies at 6 locations in the

Netherlands (Figure 1) where grassland, maize and potatoes is grown and using hydrological, soil and crop observations. The main characteristics of the 7 cases are summarized in Table 1. The soil texture ranges from sand to clay. The observations included parameters, such as groundwater levels, yields and in some cases soil moisture contents, soil pressure head and evapotranspiration. The weather data were collected from

nearby weather stations or from onsite measurements. Observations for case 1 and 2 (DM-Grass and DM-Maize) were available for a period of 22 years (1992-2013) from one field where grassland and maize was grown for respectively 7 and 15 years.

We made use of calibration work carried out by Kroes et al. (2015) and Hack et al. (2016) and limited our calibration efforts to parameter values for drought and management (Table

1), focussing on validation of results. Planting and harvest dates were given. Oxygen stress was parameterised as described by Hack et al. (2016). Drought stress was parameterised using the dry part of the reduction function proposed by Feddes et al. (1978). Drought stress is absent when the soil pressure head $h$ exceeds the critical value of $h3$. Drought stress increases linearly between $h3$ and at $h4$ (wilting point). The critical pressure head $h3$ differs

between lower and higher potential transpiration (respectively $h3l$ and $h3h$) rates.

For all cases a so-called management factor was used to close the gap between observed and actual yield. The input crop parameters for maize only differed with respect to the management factor which ranges from 0.85-0.95. The management factors are relatively high because the case study locations have good management. It is very likely that we miss

some processes even though our modelling approach is mechanistic, because it is still relatively simple. Some processes like pests and diseases are not included and may play a role in the field; the calibration was done on experimental farms where the impact from diseases and pests is minimal.

For potatoes the input crop parameters were kept the same for all 3 cases. Maximum

rooting depth for grassland, maize and potatoes were respectively 40, 100 and 50 cm.

Soil water conditions were different for all locations and boundary conditions varied, depending on local situation and available data (Table 1). In most cases a Cauchy bottom boundary condition was applied using a hydraulic head based on piezometer observations from the Dutch Geological Survey ( https://www.dinoloket.nl/). Observed groundwater levels

were used as lower boundary condition for Borgerswold (crop: potato). In 2 cases a lateral



boundary condition was applied with drainage to a surface water system (Table 1). The simulation results were analysed using an R-package (Bigiarini, 2013) and the statistics are presented in Table 2.


*2.3 Soil crop experiment to analyse the role of capillary rise*

To analyse the impact of soil type on upward soil water flow we modelled soil-crop experiments using 72 soils derived from a national soil data base (Wösten et al., 2013a). The 72 soils were aggregated from 315 soil units of the 1:50000 Dutch Soil Map using soil
hydraulic clustering methods and considering the following properties: maximum groundwater depth, saturation deficit between a certain depth and the soil surface, transmissivity for horizontal water flow, resistance for vertical water flow and availability of water in the root zone (Wösten et al., 2013b). The resulting soil hydraulic properties were subsequently used as SWAP-WOFOST input. The bottom of the soil profile is set to 5.5
meter below the soil surface. At this depth, the simulated root zone soil water fluxes are not affected anymore by the actual depth of the soil profile bottom. The bottom of the root zone is dynamic, depends on root growth and consequently varies in time.

For each soil we applied 3 hydrological conditions (Figure 2), ranging from relatively dry (a)
to relatively wet (c) The latter is the natural situation in most of the Netherlands. This hydrological condition has a fluctuating groundwater level derived from a national study (Van Bakel et al., 2008). This national study used simulation units which are unique in land use, crop type and drainage conditions resulting in daily groundwater fluctuations Lateral infiltration and drainage are accounted for ($q_{infiltration}$ and $q_{drainage}$ in Figure 2 c). We selected
three large simulation units with long term average groundwater levels between 40 and 120 cm below the soil surface (Dutch groundwater class IV, units 2245, 3859 and 621 for grassland, maize and potato, covering respectively 1806, 794 and 58102 ha, data from Van Bakel et al., 2008). See also the supplementary material of Kroes and Supit (2011) for an additional explanation of the study from Van Bakel et al (2008).
The other two conditions are unsaturated and have no groundwater due to a free-draining bottom boundary ($q_{leaching}$, see Figure 2, conditions a and b). Condition (a) has been included to explicitly demonstrate the role of upward flow. A synthetic modelling option has been implemented to stop upward flow from reaching the root zone, without inhibiting percolation. This option is implemented in the numerical solution of the Richards equation and minimizes
vertical conductivity just below the root zone in situations that the model simulates upward vertical flow. Adjustment of the code was necessary to carry out the model experiment (no recirculation) and to demonstrate (quantitatively) the added value of simulating more detailed water fluxes in the soil profile in comparison to simple bucket approaches that





inherently include the mentioned 'artificial restriction'. When crop models are used for yield
forecasting these detailed processes play an important role; neglecting them generally may
cause large errors. We want to improve our understanding of processes in the soil-crop
continuum and thereby minimizing errors. This synthetic option will be made public with the
latest release of the model.

The upward flux across the bottom of the root zone can either stem from capillary rise or
from percolation water that is recirculated ($q_{recirc}$ and $q_{caprise}$, see Figure 2 conditions b and
c). The upward flux capillary rise originates from groundwater and from recirculated
percolation water. In all hydrological conditions percolation across the root zone and
leaching across the lower boundary of the model profile occurs ($q_{percolation}$ and $q_{leaching}$ in
Figure 2). All fluxes are calculated using small variable time steps (< 1 day); however results
are accumulated to daily net fluxes, which implies that small variations within a day cannot
be seen from the results. Recirculation depends on crop water demand, soil hydraulic
properties and presence of soil moisture.

The crop parameters were kept the same as for the case studies, with a few exceptions: i)
for grassland an average management factor of 0.9 was used, ii) timing of grass-mowing
was done when a dry matter threshold of 4200 kg.ha$^{-1}$ DM was exceeded, iii) for maize and
potatoes the harvesting dates were respectively set to 25-Oct and 15-Oct.

The 3 crops and 3 lower boundary conditions resulted in 9 combinations. Each combination
was simulated with 72 soils for a period of 45 years (1971-2015) with meteorological data
from the station De Bilt (KNMI, 2016). In a subsequent analysis we grouped the results of
these 72 soils to 5 main soil groups clay, loam, peat, peat-moor and sand (Figure 3) to be
able to analyse the impact at grouped soil types.

## 3   Results

### 3.1 Case studies for validation

The first 2 case studies are from one location (De Marke) where a grassland-maize rotation
was practised. The results show that the hydrological conditions (Figure 4 and Table 2) were
simulated accurately for those years for which observed data were available (1991-1995).
From 1995-1997 the groundwater levels drop as a result of low precipitation (about 700
mm/year). The fall of the year 1998 shows rising groundwater levels that correspond well
with very wet conditions at that moment. The simulated grassland yields are overestimated
by 679 kg.ha$^{-1}$ DM and the simulated maize yields are underestimated by 285 kg.ha$^{-1}$ DM
which differences are well within acceptable ranges (Figure 5 and Table 2).



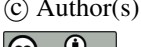

For the other 2 maize case studies (C-Maize and D-Maize) groundwater levels and soil moisture are well simulated (Table 2). The simulated maize yields (Table 2) are less acceptable for case C-Maize as is indicated by a zero or negative Nash-Sutcliffe efficiency (NS) which suggests that the observed mean is a better predictor than the model. In 1976, a

very dry year, the soil hydrology dynamics and the resulting yield were well captured. The yield of case study D-Maize has a small bias of about 300 kg.ha$^{-1}$ DM between observed and simulated.

The simulated hydrological conditions for the 3 fields of the potato-cases R-Potato and V-Potato show a good fit with the observed (Table 2). The simulated yields (Table 2) show the

largest deviation from the observed for case B-Potato. The more recent experiments of potato cases studies R-Potato and V-Potato show differences between simulated and observed yields of respectively 1400 and 300 kg.ha$^{-1}$ DM (Table 2). These case studies unfortunately cover only one year. The case R-Potato performs less due to the complex situation in the subsoil with drainage conditions that require more observations to improve

the simulations.

Even though some yields are not accurate enough to satisfy statistical criteria for good model performance, we think that the dynamics of soil hydrology and crop yield are acceptably captured. With more field information and calibration a better result could be achieved but we think that current tuning of SWAP-WOFOST for the 3 crops allows an

application at a larger scale with various hydrological boundary conditions.

Before the analysis at a larger scale we simulated the impact of upward flow for the case studies. We carried out additional simulations without upward flow towards the root zone, using the specially programmed synthetic model option. Results of these 3 cases are given

in Table 3 for the situation with and without upward flow. This shows that suppressing upward flow lowers yields by 5, 2 and 22% respectively for a grassland, maize and potato case. The groundwater recharge was reduced with respectively 3, 5 and 94% (Table 3). In a next step we carried out a larger scale experiment to quantify this impact for different soil crop and climate conditions.


*3.2 Soil crop experiment to analyse the role of capillary rise*

The 3 crops from the case studies were simulated with 72 soils from the national database using 3 different bottom boundary conditions and 45 years with weather from 1970-2015.

Results of simulated upward flow of 45 years weather, 72 soils and 3 lower boundary conditions are summarized with mean values in Table 4. The highest values for upward flow to the root zone during crop growth were found for average groundwater conditions (Ave) with long-term mean values for grassland, maize and potatoes of respectively 191, 79 and





115 mm/year. Differences among hydrological conditions at the bottom of the root zone are
caused by differences in weather, growing season, dynamic position of the root zone and
demand of root water uptake. Even in free drainage situations ($FD_{rc}$) the upward flow to the
root zone caused by soil water recirculation can be considerable, ranging from 20 – 78 mm
long-term average (Table 4). In free-draining soils the variation of capillary rise ranges from
about 10 mm in wet and cold to 150 mm in dry and warm years with a high evaporative
demand (Figures 6, upper part). In general capillary rise is highest in loamy soils where soil
physical conditions are optimal. Especially in the presence of groundwater levels differences
in capillary rise between soils are relatively small compared to differences among years and
within one grouped soil type (Figure 7, lower part).

The upward flow is inversely related to the rooting depth: the larger the rooting depth, the
smaller the upward flow. Grassland, Potatoes and Maize have rooting depths of respectively
40, 50 and 100 cm and an upward flow of respectively 191, 115 and 79 mm per growth
season (Table 4). Note that the high value for perennial grassland is also caused by a much
longer growing season. The percolation is highest for grassland for the same reasons (Table
4). These high values are largely due to the precipitation excess during winter in The
Netherlands.

Upward seepage across the bottom boundary does not occur in the free-drainage conditions
(Figure 2 a and b). Leaching is highest (Table 4) in the synthetic free-drainage condition
without capillary rise (Figure 2 a). Note that the values in Table 4 for seepage and leaching
are given for a calendar year whereas the other mean values are given for a growing
season. Yearly values are used for the bottom boundary because these values give an
indication for the yearly deeper groundwater recharge which may also be influenced by
variations of vertical fluxes close to the rootable zone during the remainder of the year. The
leaching flux at 5.5 m depth increases when upward flow is suppressed (lower transpiration,
more groundwater recharge), with respectively 43, 1 and 16 mm.year$^{-1}$ for grassland, maize
and potatoes. In Dutch conditions with shallow groundwater (Figure 2 c) very often at
greater depth leaching does not occur because excess water due to precipitation and/or
upward seepage is discharged via drainage systems. The average condition we used has
no leaching but seepage of 227, 155 and 291 mm.year$^{-1}$ for grassland, maize and potatoes.

As can be expected, the synthetic condition without upward flow and without groundwater
(Figure 2 a), has the lowest simulated mean yields for all crops (Table 4). The highest mean
yields are simulated when average groundwater situations including capillary rise are
considered (Table 4, Ave). The relative mean yield increase is lowest for maize and highest
for grassland (Table 5) which is probably caused by the difference in rooting depth.





Results of the simulations with 3 different lower boundary conditions (FD$_{nc}$, FD$_{rc}$ and Ave) are also compared by subtraction. The elimination of capillary rise to the root zone in free drainage conditions (synthetic condition a compared to b, Figure 2) reduces grassland, maize and potatoes yields with respectively 13, 1 and 8 % (Table 5). A comparison between situations with free drainage (condition b, Figure 2) with average groundwater levels (condition c, Figure 2) shows a similar yield reduction: respectively. 13, 3 and 8 %. When one compares situations with free-drainage conditions without capillary rise (synthetic condition a, Figure 2) with average groundwater levels (condition c) yield-reductions of grassland, maize and potatoes are respectively 25, 4 and 15 % (Table 5) or respectively about 3.2, 0.5 and 1.6 ton.ha$^{-1}$ dry matter (Table 4).

The impact of upward flow on groundwater recharge is highest for potatoes and lowest for maize. For grassland, maize and potatoes differences were calculated of respectively 17, -6 and 46% (Table 5) or 64, -3 and 34 mm (Table 4). Low recharge values for maize are caused by deeper rooting systems which reduce these differences because groundwater levels are closer to the bottom of the root zone. For potatoes this difference in yield can reach values of more than 4 ton.ha$^{-1}$ dry matter in stress conditions (Table 6).

The results are presented in more detail in the Supplementary Materials.

## 4. Discussion

The case studies and soil-crop experiments in this paper clearly demonstrate the combined interaction of upward flow and groundwater on crop yields. This impact is clearly present in situations where a groundwater level is present (85% of NL) but also in free-draining situations the impact of upward flow is considerable. According to our simulation results, grassland, maize and potatoes yields increase with respectively 15, 1 and 8% in free drainage conditions when upward flow is included (Table 5). This increase is mainly caused by internal recirculation, i.e. a part of the downward flux past the root zone is redirected upward to the root zone as a result of gradient driven flow. When upward flow also has groundwater as a source simulated yields increases by another 16, 3 and 9% respectively. This increase is supported by a stronger capillary rise due to proximity to the groundwater. Comparing the simple simulations (no upward flow, no groundwater influence) to those with an average groundwater level and capillary rise shows yield increases of 25, 4 and 15%. About half of these yield increases are caused by internal recirculation as occurs in free drainage conditions and the other half is caused by an increased upward capillary flow from the groundwater.





Many crop models consider the soil system as a reservoir with only percolation and no
upward flow (e.g.bucket approach). Such models do not account for soil moisture
redistribution within and below the root zone. As our simulations show, this kind of models
underestimate crop yield and overestimate groundwater recharge, and implicitly
overestimate drought stress. The irrigation demand may be overestimated as well. The high
percolation may also result in overestimation of groundwater recharge (leaching).
Groundwater depth is important, because it determines the distance that the capillary flux
has to bridge to reach the root zone and should be accounted for in crop modelling.

Our analysis shows that soil properties and soil profile layering are important because
differences in soil hydraulic properties influence vertical water flow. High upward flow values
are found in loamy soils as is expected (Table 6, max row), but if water stress is high and
upward flow is low the influence of soil type decreases. Low upward flow values were found
for loamy soils (Table 6, min row). Comparing the minimum yield values it shows that there
is a large difference between these soil types in free-drainage conditions with and without
upward flow. This means that the storage capacity of loamy soils is larger than the one of
sandy soils as can be expected. The yield variation between soil types in water stress
conditions is large and illustrates the need for a proper soil schematization especially in
stress full hydrological conditions. An adequate soil schematization is relevant for all models
but especially for those that use a bucket approach ($FD_{NC}$). As the influence of recirculation
($FD_{rc}$) increases, the yield variation becomes less and the influence of soil type decreases.
In situations without water stress the soil type is less important. In conditions where
groundwater and capillary rise occurs (Ave) yield variation is hardly influenced by soil type.
Modelling concepts should consider dynamic interactions between soil water and crop
growth. Crop models in general should consider recirculation of soil water and, especially in
low lying regions like deltas, groundwater dynamics should be considered as well.


Precipitation, soil texture and water table depth jointly affected the amount of groundwater
recharge and time-lag between water input and groundwater recharge (Ma et al., 2015). We
quantified some of these issues, but several items remain, such as the impact of rooting
depth on crop yield and transpiration. Also soil and water management practises like
ploughing and irrigation, are not considered. Furthermore the rooting pattern needs a more
detailed analysis; we applied an exponential decrease of root density and compensation of
root uptake according to Jarvis (2011) but the macroscopic root water uptake concept is still
simple and requires a more detailed analyses (Dos Santos et al. 2017). Another item we
neglected is the preferential flow of water by the occurrence of non-capillary sized
macropores (Bouma, 1961, Feddes, 1988), which is relevant in especially clay soils.
Hysteresis of the water retention function is also not considered. An additional analysis of



these issues is recommended, especially the impact of different rooting patterns on capillary rise should be addressed.

The impact of soil type on yield increases when environmental conditions become dryer; situations without groundwater and without upward flow have less yield and higher yield variation then situations where groundwater influences capillary rise (For detailed information on results see the Supplementary Materials).


*5.  Conclusions*

We quantified the impact of upward flow on crop yields of grassland, maize and potatoes in layered soils. We compared situations with average groundwater levels with free-drainage

conditions with and without upward flow. The largest difference was found when one compares situations with average groundwater levels with free drainage conditions without upward flow. From these differences one may conclude that neglecting upward flow has a large impact on simulated yields and water balance calculations especially in regions where shallow groundwater occurs. The comparison shows long term average yield-reductions of

grassland, maize and potatoes of respectively 25, 4 and 15 % (Table 5) or respectively 3.2, 0.5 and 1.6 ton Dry Matter per ha (Table 4). Reduction of the percolation flux can be considerable; for grassland and potatoes the reduction is 17 and 46% (Table 5) or 70 and 34 mm (Table 4).

About half of the yield increases is caused by internal recirculation as occurs in free-

drainage conditions and the other half is caused by an increased upward capillary flow from groundwater. Improved modelling should consider upward flow of soil water which will result in improved estimates of crop yield and percolation.

We think that the quantification of upward flow on yield is a novelty, especially with respect to the interaction between recirculation and crop growth. Studies about the relation between

soil hydrology and crop growth should quantify this upward flow because neglecting this flow and its impact implies neglecting yield changes which may have a large economic value in the Dutch Delta and in other deltas in general. Another aspect which cannot be found in the referenced studies is the lack of a quantification of the impact of capillary rise and recirculation on crop yields. Correct quantification of the water fluxes contributes to the

understanding  of crop production and will help the institutions in charge of yield forecasting.

*Acknowledgement*



Part of the case studies has been used before (Hack et al., 2016). This project is related to
the project WaterVision Agriculture (www.waterwijzer.nl) which is financed by a large group
of financers: STOWA (Applied Research of the Water Boards), Ministry of Infrastructure and
Environment, ACSG (Advisory Commission for Damage related to Groundwater), provinces
Utrecht and Zuid-Holland, ZON (Zoetwatervoorziening Oost-Nederland), Water companies
Vitens and Brabant Water, VEWIN, LTO and the Ministry of Economic Affairs (project KB-
490    14-001-046).

We also thank three anonymous reviewers for their constructive and valuable comments on
earlier versions of this paper.





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





Tables

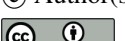



*Table 1. Main characteristics of case studies used to verify setup of model combination SWAP-WOFOST*

| Case study | Crop | Location | Period | Soil type | Observations[1] | Reference | Drought stress[2] | MF[3] | RZ[4] | BBC[5] | Lateral Boundary |
|---|---|---|---|---|---|---|---|---|---|---|---|
| DM-Grass | Grass-land | De Marke | 1995-1996, 2005-2008, 2013 | dry sandy soil | Gwl, Yield, Theta20cm | Hack et al. (1996); Verloop et al., 2014 | h3h = -200.0 cm; h3l = -800.0 cm; h4 = -8000.0 cm | 0.8 | 40 | | No Drainage |
| DM-Maize | Silage maize | De Marke | 1992-1994, 1997-2003, 2009-2012 | dry sandy soil | Gwl, Yield, Theta20cm | Hack et al. (1996); Verloop et al., 2014 | | 0.90 | | Cauchy | No Drainage |
| C-Maize | Silage maize | Cranendonck | 1974-1982 | Cumulic Anthrosol | Gwl, Yield | Schröder (1985) | h3h = -400.0 cm; h3l = -500.0 cm; h4 = -10000.0 cm | 0.85 | 100 | Cauchy | No Drainage |
| D-Maize | Silage maize | Dijkgraaf | 2007 | Umbric Gleysol | Gwl, Yield, ET, Theta20cm | Elbers et al. (2010) | | 0.95 | | Cauchy | No Drainage |
| B-Potato | Potato | Borgerswold | 1992, 1994 | Sandy loam | Gwl, Yield | Dijkstra et al., 1995 | | | | Observed groundwater | No Drainage |
| R-Potato | Potato | Rusthoeve | 2013 | lichte kleibodem | Gwl, Yield, Qdrain | Van Den Brande (2013) | h3h = -300.0 cm; h3l = -500.0 cm; h4 = -10000.0 cm | 0.8 | 50 | Cauchy | Drain tubes at -90 cm |
| V-Potato | Potato | Vredepeel | 2002 | Sandy loam | Gwl, Yield | De Vos et al., 2006 | | | | Closed | Drain ditch at -100 cm |

[1] Gwl = Groundwater level, Yield = Actual Yield as Dry Matter of Harvested product, Theta20cm= Soil moisture content at a depth of 20cm below surface, Qdrain = drainage from field to surface water via tube drains, ET = Evapotranspiration measured via Eddy Correlation method.
[2] h3h = h below which water uptake reduction starts at high Tpot; h3l = h below which water uptake red. starts at low Tpot; h4 = No water extraction at lower pressure heads; Drought stress was parameterised using the dry part of the reduction function proposed by Feddes et al. (1978), Drought stress is absent when the soil pressure head h exceeds the critical value of h3. Drought stress increases linearly between h3 and at h4 (wilting point). The critical pressure head h3 differs between lower and higher potential transpiration (Tpot) (respectively h3l and h3h) rates. [3] MF = Management Factor to account for imperfect management
[4] RZ = Maximum depth of root zone (cm)
[5] BBC = Bottom Boundary Condition. The Cauchy bottom boundary condition uses a hydraulic head based on piezometer observations from an open data portal (see text)








*Table 2. Results of Case studies: simulated and observed values*

| Case study | Name[0] | unit | Simulated mean | Observed mean | ME[1] | RMSE[2] | NS[3] | d[4] | n[5] |
|---|---|---|---|---|---|---|---|---|---|
| DM-Grass | Yield | kg.ha$^{-1}$.yr$^{-1}$ DM | 11728 | 11049 | 679 | 1343 | 0.6 | 0.9 | 7 |
| | Gwl | m-soil | -1.34 | -1.30 | -0.04 | 0.46 | 0.3 | 0.9 | 77 |
| | Theta20cm | m$^3$.m$^{-3}$ | 0.27 | 0.27 | 0.01 | 0.06 | 0.5 | 0.9 | 43 |
| DM-Maize | Yield | kg.ha$^{-1}$.yr$^{-1}$ DM | 11564 | 11850 | -286 | 2825 | -3.2 | 0.4 | 14 |
| C-Maize | Yield | kg.ha$^{-1}$.yr$^{-1}$ DM | 14054 | 13788 | 266 | 2587 | -1.1 | 0.7 | 9 |
| | Gwl | m-soil | -1.42 | -1.36 | -0.05 | 0.25 | 0.4 | 0.9 | 61 |
| D-Maize | Yield | kg.ha$^{-1}$.yr$^{-1}$ DM | 15974 | 16306 | -332 | | | | 1 |
| | LAI | m$^2$.m$^{-2}$ | 2.1 | 2.5 | -0.3 | 0.6 | 0.7 | 0.9 | 10 |
| | ETact | mm.yr$^{-1}$ | 1.4 | 1.9 | -0.6 | 0.9 | 0.5 | 0.9 | 232 |
| | Gwl | m-soil | -1.03 | -1.07 | 0.03 | 0.06 | 0.9 | 1.0 | 112 |
| | Theta20cm | m$^3$.m$^{-3}$ | 0.29 | 0.27 | 0.01 | 0.03 | 0.5 | 0.8 | 219 |
| B-Potato | Yield | kg.ha$^{-1}$.yr$^{-1}$ DM | 10532 | 9246 | 1286 | 1350 | -31.9 | 0.3 | 2 |
| | Gwl | m-soil | -1.10 | -1.10 | 0.00 | 0.03 | 1.0 | 1.0 | 123 |
| R-Potato | Yield | kg.ha$^{-1}$.yr$^{-1}$ DM | 10019 | 8610 | -1409 | | | | 1 |
| | Gwl | m-soil | -1.07 | -1.10 | 0.02 | 0.19 | 0.6 | 0.9 | 887 |
| | qDrain | mm | 1.1 | 0.6 | 0.4 | 1.4 | 0.4 | 0.8 | 1084 |
| V-Potato | Yield | kg.ha$^{-1}$.yr$^{-1}$ DM | 11071 | 11359 | -288 | | | | 1 |
| | Gwl | m-soil | -1.04 | -1.07 | 0.03 | 0.11 | 0.8 | 0.9 | 353 |

[1] *Gwl = Ground Water Level; Theta20cm = Volumic Soil Moisture Content at a depth of 20 cm below the soil surface; LAI=Leaf Area Index; ETact = actual EvapoTranspiration; qDrain = Drainage flux*

[1] *ME: Mean Error between simulated (sim) and observed (obs), in the same units of sim and obs, with treatment of missing values. A smaller value indicates better model performance*

[2] *RMSE: Root Mean Square Error between sim and obs, in the same units of sim and obs, with treatment of missing values. RMSE gives the standard deviation of the model prediction error. A smaller value indicates better model performance.*

[3] *NS: Nash-Sutcliffe efficiencies range from -Inf to 1. Essentially, the closer to 1, the more accurate the model is. NS = 1, corresponds to a perfect match of modelled to the observed data. NS = 0, indicates that the model predictions are as accurate as the mean of the observed data. -Inf < NS < 0, indicates that the observed mean is better predictor than the model.*

[4] *d: The Index of Agreement (d) developed by as a standardized measure of the degree of model prediction error*
*and varies between 0 and 1. A value of 1 indicates a perfect match, and 0 indicates no agreement at all. The index of agreement can detect additive and proportional differences in the observed and simulated means and variances; however, it is overly sensitive to extreme values due to the squared differences.;*

[5] *n: the number of values used with the previous 4 statistical criteria to compare simulated and observed results.*




*Table 3. Results of case studies: values and differences of yield, capillary rise and percolation fluxes, resulting from simulations with and without capillary rise*

| Case study[1] | Model Result | Condition A[2] | B[3] | Differences A-B | Unit | Differences (%) 100*(A-B)/A |
|---|---|---|---|---|---|---|
| DM-Grass | $Y_{act}$ | 11904 | 11353 | 551 | kg.ha$^{-1}$. season$^{-1}$ DM | 5 |
| | $q_{caprise}$ | 28 | 0 | 28 | mm.season$^{-1}$ | 100 |
| | $q_{percolation}$ | 314 | 306 | 9 | mm.season$^{-1}$ | 3 |
| DM-Maize | $Y_{act}$ | 12504 | 12257 | 247 | kg.ha$^{-1}$. season$^{-1}$ DM | 2 |
| | $q_{caprise}$ | 8 | 0 | 8 | mm.season$^{-1}$ | 100 |
| | $q_{percolation}$ | 81 | 77 | 4 | mm.season$^{-1}$ | 5 |
| V-Potato | $Y_{act}$ | 11071 | 8665 | 2406 | kg.ha$^{-1}$. season$^{-1}$ DM | 22 |
| | $q_{caprise}$ | 105 | 0 | 105 | mm.season$^{-1}$ | 100 |
| | $q_{percolation}$ | 15 | 1 | 14 | mm.season$^{-1}$ | 94 |

[1] *Cases studies DM-Grass and DM=Maize were simulated for limited periods of respectively 2005-*
*2008 and 1991-1994 to have a continuous sequence of years, Case study V-Potato was simulated for one year*
[2] *Condition A has actual bottom boundary conditions (according to table 1);*
[3] *Condition B has actual bottom boundary conditions (table 1) but without capillary rise to root zone;*


*Table 4. Results of soil crop experiments: mean values of 6 model results from 3 different hydrological conditions ($FD_{nc}$, $FD_{rc}$ and Ave)*

| Crop | Model Result | $FD_{nc}$ | $FD_{rc}$ | Ave | Unit |
|---|---|---|---|---|---|
| Grassland | $Y_{act}$ | 9781 | 11227 | 12978 | kg.ha$^{-1}$.season$^{-1}$ DM |
| | $q_{caprise}$ | 0 | | 191 | mm.season$^{-1}$ |
| | $q_{recirc}$ | 0 | 78 | | mm.season$^{-1}$ |
| | $q_{percolation}$ | 318 | 340 | 382 | mm.season$^{-1}$ |
| | $q_{seepage}$ | 0 | 0 | 227 | mm.yr$^{-1}$ |
| | $q_{leaching}$ | 302 | 259 | 0 | mm.yr$^{-1}$ |
| Maize | $Y_{act}$ | 12127 | 12239 | 12626 | kg.ha$^{-1}$.season$^{-1}$ DM |
| | $q_{caprise}$ | 0 | | 79 | mm.season$^{-1}$ |
| | $q_{recirc}$ | 0 | 20 | | mm.season$^{-1}$ |
| | $q_{percolation}$ | 48 | 53 | 45 | mm.season$^{-1}$ |
| | $q_{seepage}$ | 0 | 0 | 155 | mm.yr$^{-1}$ |
| | $q_{leaching}$ | 381 | 380 | 0 | mm.yr$^{-1}$ |
| Potato | $Y_{act}$ | 8764 | 9482 | 10342 | kg.ha$^{-1}$.season$^{-1}$ DM |
| | $q_{caprise}$ | 0 | | 115 | mm.season$^{-1}$ |
| | $q_{recirc}$ | 0 | 44 | | mm.season$^{-1}$ |
| | $q_{percolation}$ | 39 | 50 | 73 | mm.season$^{-1}$ |
| | $q_{seepage}$ | 0 | 0 | 291 | mm.yr$^{-1}$ |
| | $q_{leaching}$ | 423 | 407 | 0 | mm.yr$^{-1}$ |




*Table 5. Results of soil crop experiments: differences (%) between results from 3 different hydrological conditions ($FD_{nc}$, $FD_{rc}$ and Ave)*

| crop | model Result | differences (%) | | |
|---|---|---|---|---|
| | | $100*(FD_{rc} - FD_{nc}) / FD_{rc}$ | $100*(Ave- FD_{rc}) / Ave$ | $100*(Ave- FD_{nc}) / Ave$ |
| Grassland | $Y_{act}$ | 13 | 13 | 25 |
| | $q_{percolation}$ | 6 | 11 | 17 |
| Maize | $Y_{act}$ | 1 | 3 | 4 |
| | $q_{percolation}$ | 10 | -18 | -6 |
| Potato | $Y_{act}$ | 8 | 8 | 15 |
| | $q_{percolation}$ | 22 | 32 | 46 |

*Table 6. Results for potatoes of soil crop experiments for each clustered soil type: capillary rise, recirculation and yield from 3 different hydrological conditions ($FD_{nc}$, $FD_{rc}$ and Ave). Results for upward flow rise of $FD_{nc}$ are zero and therefore not given.*

| hydrological condition | | Statistic | Values per clustered soil type | | | | | Unit |
|---|---|---|---|---|---|---|---|---|
| | | | Clay | Loam | Peat | Moor | Sand | |
| $FD_{rc}$ | $q_{recirc}$ | min | 6 | 2 | 4 | 9 | 1 | mm/crop season |
| | | lower quartile | 35 | 35 | 21 | 28 | 26 | mm/crop season |
| | | median | 49 | 56 | 36 | 35 | 38 | mm/crop season |
| | | upper quartile | 64 | 80 | 52 | 41 | 53 | mm/crop season |
| | | max | 106 | 124 | 93 | 58 | 91 | mm/crop season |
| Ave | $q_{caprise}$ | min | 15 | 14 | 16 | 35 | 15 | mm/crop season |
| | | lower quartile | 67 | 83 | 75 | 103 | 86 | mm/crop season |
| | | median | 97 | 116 | 107 | 132 | 116 | mm/crop season |
| | | upper quartile | 137 | 155 | 144 | 171 | 155 | mm/crop season |
| | | max | 231 | 241 | 235 | 246 | 253 | mm/crop season |
| $FD_{nc}$ | $Y_{act}$ | min | 2.9 | 5.3 | 2.7 | 2.6 | 1.2 | 1000 kg/ha DM |
| | | lower quartile | 7.3 | 8.6 | 7.5 | 6.8 | 6.7 | 1000 kg/ha DM |
| | | median | 9.5 | 10.2 | 9.7 | 9.2 | 9.0 | 1000 kg/ha DM |
| | | upper quartile | 10.7 | 10.9 | 10.7 | 10.6 | 10.7 | 1000 kg/ha DM |
| | | max | 12.2 | 12.2 | 12.2 | 12.2 | 12.2 | 1000 kg/ha DM |
| $FD_{rc}$ | $Y_{act}$ | min | 5.1 | 7.4 | 4.7 | 3.3 | 3.1 | 1000 kg/ha DM |
| | | lower quartile | 8.5 | 9.6 | 8.4 | 7.5 | 7.7 | 1000 kg/ha DM |
| | | median | 10.1 | 10.6 | 10.1 | 9.7 | 9.8 | 1000 kg/ha DM |
| | | upper quartile | 10.9 | 11.1 | 10.9 | 10.8 | 10.8 | 1000 kg/ha DM |
| | | max | 12.4 | 12.4 | 12.4 | 12.4 | 12.4 | 1000 kg/ha DM |
| Ave | $Y_{act}$ | min | 7.3 | 7.7 | 7.8 | 7.8 | 7.6 | 1000 kg/ha DM |
| | | lower quartile | 9.6 | 9.8 | 9.8 | 9.8 | 9.7 | 1000 kg/ha DM |
| | | median | 10.5 | 10.7 | 10.7 | 10.7 | 10.7 | 1000 kg/ha DM |
| | | upper quartile | 11.1 | 11.2 | 11.1 | 11.2 | 11.1 | 1000 kg/ha DM |
| | | max | 12.6 | 12.6 | 12.6 | 12.6 | 12.6 | 1000 kg/ha DM |



## Figures


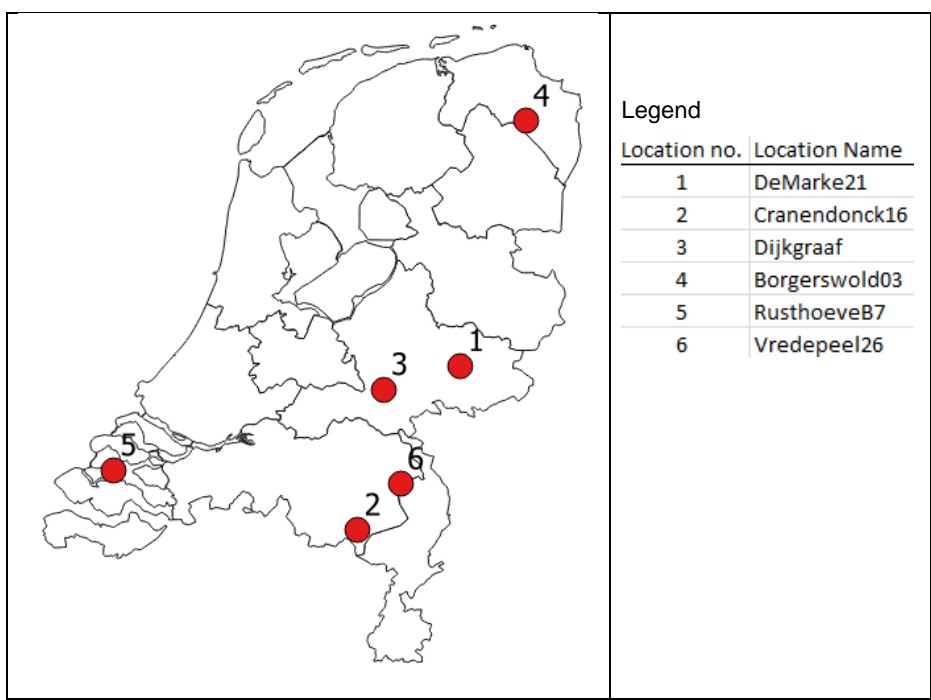

*Figure 1. Location of case studies for grassland, maize and potatoes; location De Marke has a rotation of grassland and maize on the same field.*





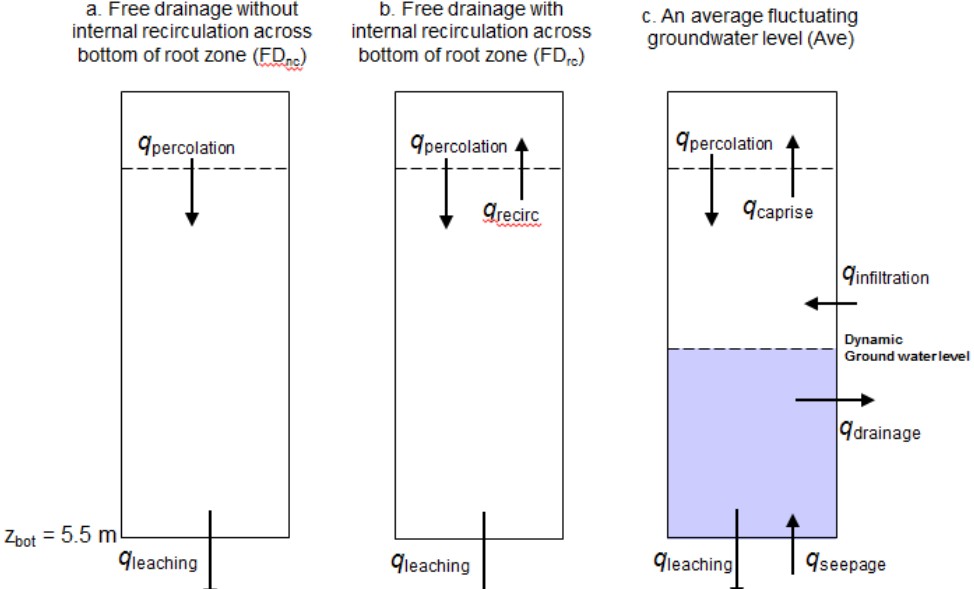

*Figure 2. Schematization of 3 hydrological conditions: a. Free Drainage without recirculation across bottom of rootzone ($FD_{nc}$), b. Free Drainage with recirculation across bottom of rootzone ($FD_{rc}$) and c. Average fluctuating groundwater level (Ave).*
*Conditions a and b have free-draining bottom boundary conditions without groundwater. Condition a is artificially created to explicitly demonstrate the role of recirculating percolation resulting in upward flow to the rootzone. Condition b is a common free drainage situation which includes upward flow due to recirculating percolation water. Condition c is the natural situation in most of the Netherlands. This hydrological condition has a fluctuating groundwater level derived from a national study (Van Bakel et al., 2008).*





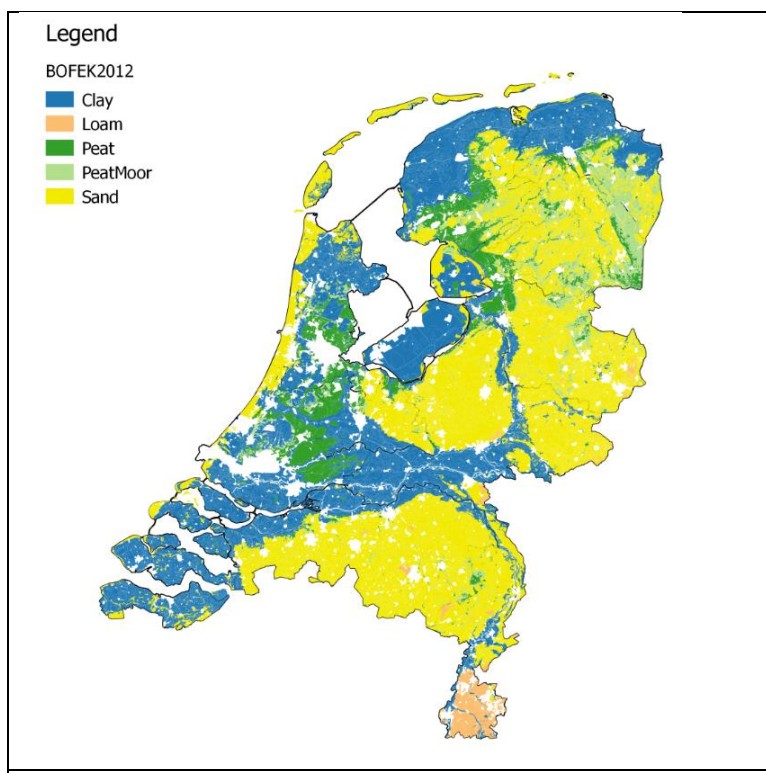

Figure 3. Five grouped soil types, based on 72 soils of the Soil Physical Map of The Netherlands (Wösten et al., 2013)





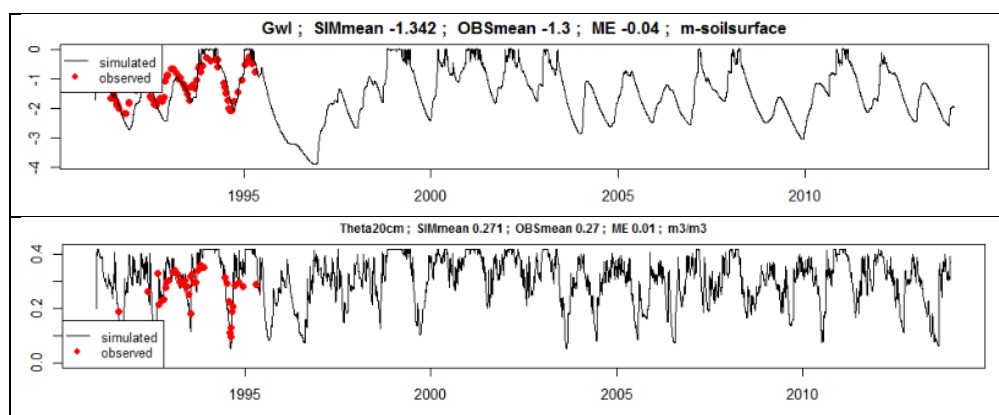

Figure 4. Results of case studies for grassland at location 1 (De Marke):
-    top figure = groundwater level (Gwl in m-soil surface):
-    bottom figure = soil moisture content (Theta20cm in $m^3.m^{-3}$) at 20 cm below the soil
     surface


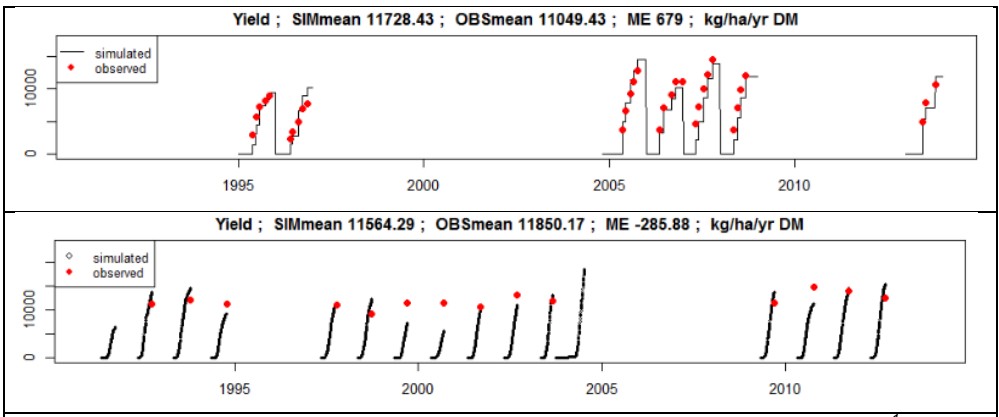

Figure 5. Results of case studies at location 1 (De Marke) : Observed Yields (kg.ha$^{-1}$DM) as
red dots and Simulated above ground biomass as black lines or as black dots
-    top figure = yields (kg.ha$^{-1}$ DM) of grassland
-    bottom figure = yields (kg.ha$^{-1}$ DM) of maize






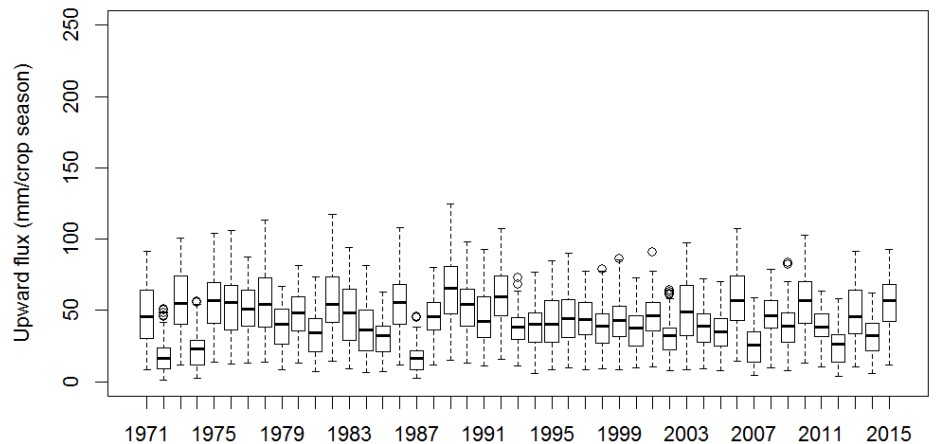

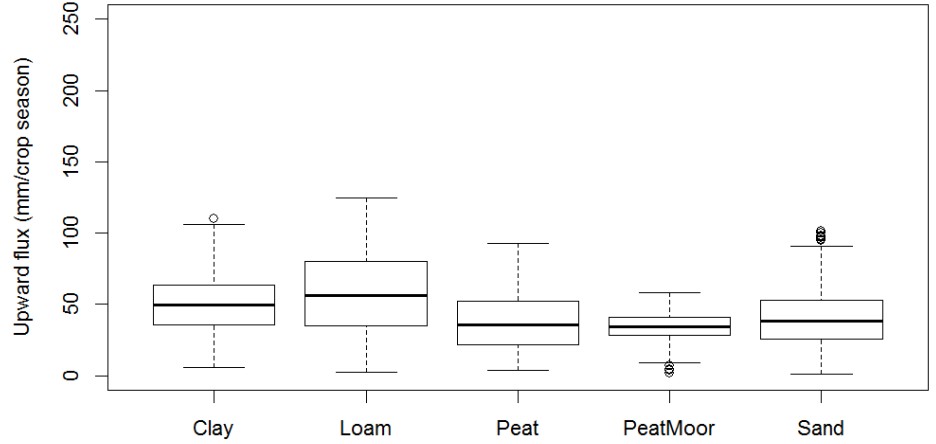

*Figure 6. Results of soil-crop experiment for potato: Upward flux across the bottom of the rootzone*
*($q_{recirc}$ in mm.crop season$^{-1}$) for hydrological conditions with free drainage ($FD_{rc}$);*
*Upper figures: results for all 72 soils for the period 1971-2015;*
*Lower figures: results as boxplots for clustered soil types.*



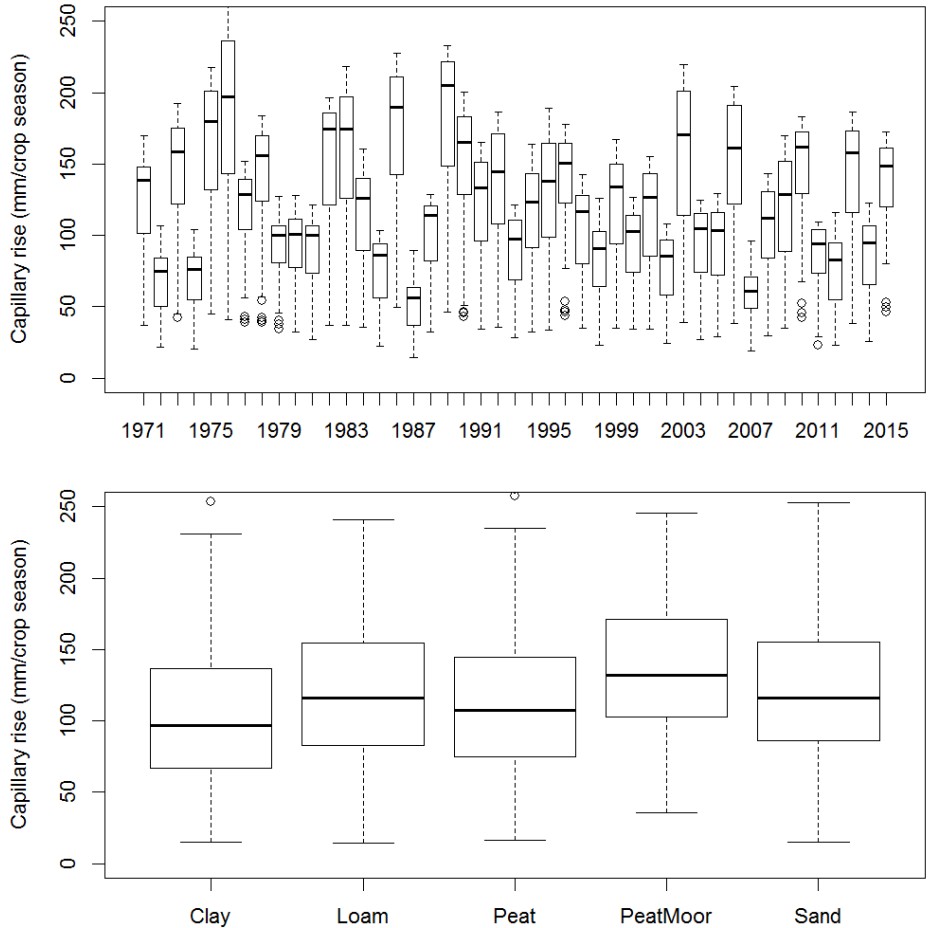

Figure 7. Results of soil-crop experiment for potato: Upward flux across the bottom of the rootzone
($q_{caprise}$ in mm.crop season$^{-1}$) for hydrological conditions with average groundwater level (Ave);
Upper figures: results for all 72 soils for the period 1971-2015;
Lower figures: results as boxplots for clustered soil types.
