# Peer review of "Impact of capillary rise and recirculation on simulated crop yields"

_Hydrology and Earth System Sciences, 2017_

## Referee Comment (RC1) · Anonymous Referee #1 · 24 Apr 2017

Comments on "Impact of capillary rise and recirculation on crop yields" by Joop Kroes, Iwan Supit, Jos van Dam, Paul van Walsum, and Martin Mulder. Wageningen University and Research, Environmental Research (Alterra) (JK; IS; PvW; MM); Wageningen University and Research – Chair Water Systems and Global Change (IS); and Wageningen University and Research – Chair Soil Physics and Land Management (JvD). Hydrology and Earth System Sciences Manuscript No. hess-2017-223. Review done April 23-24, 2017.

I am not a modeler, so I can comment on this paper only in a general way.

The authors acknowledge three anonymous reviewers of an earlier version of this manuscript. Because apparently the authors have revised the paper according to their comments, the paper should be in good shape. In addition, one of the authors (J. van

Dam) is a well-known soil physicist and has published in the literature since at least 1992.

General comments:

In the Introduction, the authors say, "...however, we found only a few studies...to quantify capillary rise...using physically based approaches" (see I. 91-94). In the Conclusions, they say that their "quantification of upward flow on yield is a novelty" (see I. 473). They also say, "Another aspect which cannot be found in the referenced studies is the lack of a quantification of the impact of capillary rise and recirculation on crop yields" (see I. 477-479). The authors are ignoring the work of the early Dutch physical scientist, Symen Barend Hooghoudt. He was famous for developing the theory for the flow of water to ditches and drains in the shallow soils of the Netherlands. See the following biography of him:

Raats, P.A.C., and R.R. van der Ploeg. 2005. Hooghoudt, Syman Barend, p. 188-195. In: D. Hillel (Editor). Encyclopedia of Soils in the Environment. Vol. 2. Elsevier, Amsterdam.

Hooghoudt modified the ellipse equation for equally spaced drainage ditches overlying an impervious layer. He wrote mainly in Dutch. In one publication that I have, he quantifies capillary rise. See:

Hooghoudt, S.B. 1937. Bijdragen Tot de Kennis van Eenige Natuurkundige Groot-Heden van den Gond. 6. Bepaling van de Doorlatendheid in Gronden van de Tweede Soort; Theorie en Toepassingen van de Kwantitatieve Strooming van het Water in Ondiep Gelegen Grondlagen, Vooral in Verband met Ontwaterings- en Infiltratievraagstukken. Departement van Economische Zaken Directie van den Landbouw. Verslagen van Landbouwkundige Onderzoekingen No. 43 (13) B, p. 461-676. Bodemkundig Instituut te Groningen. Rijksuitgeverij Dienst van de Nederlandsche Staatscourant. 'S-Gravenhage, Algemeene Landsdrukkerij. HESSD
He has a paper in English in which he talks about capillary rise, ground-water level, and crop yield. See:

Hooghoudt, S.B. 1952. Tile drainage and subirrigation. Soil Science 74:35-48.

In this paper, see his Figure 4, where he plots the yield of potatoes versus ground-water level. He considers both arable land and grassland, and he points out that grassland requires less drainage than arable land.

I think the authors should recognize that quantitative work was done on capillary rise and crop yields by Hooghoudt, which was long before computer models were used.

I do not understand Figure 2b. The authors show no impervious layer at the bottom of the figure. So how can water move upward by "recirculation"? Without an impervious layer, it seems to me that Figure 2b should be the same as Figure 2a.

Specific comments:

I. 42 and I. 623: This should be "SSSA," not "SSA." The name of the society is the Soil Science Society of America (SSSA).

I. 52-54: Can the authors give the common names of these soils? Are they sandy soils? Clayey soils?

I. 57, 69, 72, and 125: Give the scientific name of the plant along with the common name (maize, quinoa, soybean, and potatoes).

I. 96: What does "groundwater yield subsidy ss" mean? What does the "ss" stand for? What are the units for "groundwater yield subsidy"?

I. 101: "the difference in soil water potential"—difference between what? The authors need to be specific in their definitions.

I. 106: This should be "Richards' equation," not "Richard's equation." The name of the person is L.A. Richards. The authors have written this term in three different ways: as
here (I. 106); as "Richards' equation" (I. 143); and as "Richards equation" (I. 254). It is usually written as "Richards equation." The editions of Soil Physics write it as Richards equation. For example, see:

Jury, W.A., W.R. Gardner, and W.H. Gardner. 1991. Soil Physics. Fifth edition. Wiley, New York. 328 pp.

I. 109: SWAP should be defined the first time it is used (here). It is not defined until I. 160 (soil-water-atmosphere-plant).

I. 109 and 138: What does WOFOST stand for? Please write it out.

I. 153: It should be "van Genuchten" (no capital letter on the "v" in "van").

I. 179: Put the "2" in CO2 as a subscript.

I. 190: Change "is grown" to "are grown" ("...grassland, maize and potatoes are grown...").

I. 195-196: Define DM.

I. 214: What "3 cases"? I do not see where these three cases have been defined in the text so far. The authors refer to "7 case studies" on I. 189.

I. 246: I do not understand what "units 2245, 3859, and 621" mean. Please define these numbers.

I. 259: I do not see where the term "artificial restriction" has been used previously in the text. Can the authors point out where it has been used?

I. 275: I assume that here DM stands for "dry matter." Is this what DM stands for on I. 195-196?

I. 296, 305, and 306: What do C, D, B, R, and V stand for?

I. 336 (here FDrc) and elsewhere in the text: At the beginning of the paper, I suggest that the authors have a list of abbreviations with units for each parameter, so the reader
knows what the abbreviations stand for.

I. 345: Delete the capital letters on "potatoes" and "maize."

I. 349-350: The authors here for the first time in the text capitalize "The" on "The Netherlands." Previously, they have written it "the Netherlands" (e.g., see I. 75). Be consistent in writing the name of the country.

I. 372: As noted above, the authors need a list of abbreviations. I had forgotten what "Ave" stands for and had to search back in the text to find its meaning (see I. 332—"average groundwater conditions").

I. 383: "differences" —what differences? Difference between what and what?

I. 393, 394: Delete "clearly." Do not editorialize. This may not be clear to some readers.

I. 421-422: "Low upward flow values were found for loamy soils..." This appears to contradict what the authors say in I. 84-86, as follows: "Rijtema (1971) estimated that loamy soils have an almost 2 times higher capillary rise than sandy soils." Can the authors please clarify these seemingly contradictory statements?

I. 430: Soil type is important in determining capillary rise. See Figure 46 in Tolman's book, where he shows rate and extent of capillary rise in five different soils (sand, clay, clay loam, fine sandy loam, and sandy loam). The reference is:

Tolman, C.F. 1937. Ground Water. McGraw-Hill Book Co., Inc., New York. 593. See Fig. 46 on p. 157.

Tolman has another figure on p. 157 (Fig. 47) showing that the height of capillary fringe is higher in a subsiding (falling) water table than in a rising water table. This is because super-capillary sized pores are filled in a falling water table, but they are empty in a rising water table. The authors have not considered the difference in amount of water in the capillary fringe in a rising or falling water table (hysteresis). The authors say this on I. 446.
I. 452: Change "then" to "than" ("...higher yield variation than situations...").

I. 460: "The largest difference"Tof what? Some people read only the "Conclusions" of a paper, so everything should be defined in the concluding section.

I. 494, References: Make sure the references are in a common format. For example, sometimes the authors put the year in parentheses and sometimes they do not (e.g., compare I. 515 and I. 518).

Tables: Each table and figure should be self-explanatory, so all abbreviations should be defined.

In Table 1, define DM, C, D, B, R, and V. In Tables 5 and 6, define FDnc, FDrc, and Ave.

Figures 4, 5, 6, and 7: The orientation of the numbers on the y-axis is incorrect. The numbers need to be rotated 90o to the left. The numbers need to face the reader straight on.

---

## Author Comment (AC1) · 5 Jul 2017

Reviewer's General comment 1: "The authors are ignoring the work of the early Dutch physical scientist, Symen Barend Hooghoudt" Our reply to General comment 1: As the reviewer suggested we read [1], [2] and [3]. We agree with the reviewer that Hooghoudt did quantative work on capillary rise and crop yields, especially in [1]. Indeed Hooghoudt mentions the relation between capillary layers and yields in at least 2 publications mentioned by the reviewer. But his relations are always related to groundwater tables. Recirculation as we discuss is not mentioned. However the quantitative work done by Hooghoudt on capillary rise and crop yields was long before computer models, it should be recognised and we will refer to his early work when we talk about the importance of capillary rise and refer to his earliest publication [1] in line 90. Ref-

erences: [1] Hooghoudt, S.B., 1937. Contributions to the Knowledge of Some Physical Soil Properties. No. 43 (13) B, p. 461-676. Determination of the Conductivity of Soils of the Second Kind (In Dutch: Verslagen van Landbouwkundige Onderzoekingen, no. 43(13) B, Dep. van Economische Zaken, Directie van den Landbouw. Algemeene Landsdrukkerij), The Hague. [2] Hooghoudt, S. B. (1952). Tile drainage and subirrigation. Soil Science, 74(1), 35–48. [3] Raats, P. A. C., & Ploeg, R. R. van der. (2005). Hooghoudt, Syman Barend. In D. Hillel (Ed.), Encyclopedia of Soils in the Environment. Vol. 2 (pp. 188–195). Elsevier, Amsterdam.

Reviewer's General comment 2 : "I do not understand Figure 2b." Our reply to General comment 2: Figure 2b illustrates the hydrological condition "Free Drainage with recirculation across bottom of Rootzone". As stated below the figure this condition b is a common free drainage situation which includes upward flow due to recirculating percolation water. Recirculation also happens in free drainage situations. We quantified recirculation separately from capillary rise using model experiments illustrated in figure 2a and 2b. see also the special paragraph in lines 250-264 To make the role of recirculation even more explicit we changed line 101 from "The driving force for induced capillary rise is the difference in soil water potential," into "The driving force for induced capillary rise "and recirculation" is the difference in soil water potential,"

Specific comment From reviewer: l. 42 and l. 623: Our reply: SSA was changed into SSSA

Specific comment From reviewer: l. 52-54: Our reply: The use of names of USDA soil taxonomy can be avoided here. Aquepts were explained because this is relevant. The sentence with names of Histosols and Inceptisols was deleted because it has very little added value.

Specific comment From reviewer: l. 57, 69, 72, and 125: Our reply: We verified the references and give both scientific and common name for those crops where the reference also uses Scientific names. This was the case in l.69 (quinoa) and l.72 (coybean)

where we added the scientific name according to the reference. This comment also made us look at the reference Geerts et al (2006) which was missing in the References and therefore added to the References.

Specific comment From reviewer: l. 96: Our reply: i) the "ss" was a typing error and is changed into "as" ii) Zipper et al.(2015) quantified "groundwater yield subsidy" as harvested biomass in Mg.ha-1 Dry Matter. The unit was added to the text.

Specific comment From reviewer: l. 101: Our reply: clarified by defining the difference ( =" at different soil depths")

Specific comment From reviewer: l. 106: Our reply: "Richards equation" was applied throughout the text

Specific comment From reviewer: l. 109: Our reply: we defined SWAP at the suggested location. And added the latest reference to the model: Kroes et al. (2017).

Specific comment From reviewer: l. 109 and 138: Our reply: WOFOST is defined at the first place where it is mentioned, similar to SWAP.

Specific comment From reviewer: l. 153: Our reply" "van Genuchten" (no capital letter, but "v" in "van") was applied

Specific comment From reviewer: l. 179: Our reply: CO2 as $CO_2$

Specific comment From reviewer: l. 190: Our reply: "is grown" changed to "are grown" and the sentence was improved.

Specific comment From reviewer: l. 195-196: Our reply: Table 1 is extended with an explanation of the acronyms used for the names of the case studies.

Specific comment From reviewer: l. 214: Our reply: by referring to Table 1 this should become clear.

Specific comment From reviewer: l. 246: Our reply: this is too much detail, and also explained in the extended material. So we deleted part of the sentence between brackets and referred to the extended material(s).

Specific comment From reviewer: l. 259: Our reply: 'artificial restriction' is not mentioned before, se we deleted the phrase "that inherently include the mentioned 'artificial restriction'"

Specific comment From reviewer: l. 275: Our reply: DM indeed stands for Dry Matter. We agree that it was confusing. But with the explanation given in Table 1 we hope that this is clear now (see also our previous reply to comments: l. 195-196). DM is used 47 times, the first wime DM is used as Dry Matter we give now an explanation.

Specific comment From reviewer: l. 296, 305, and 306: Our reply: These acronyms are now explained in Table 1 (see also our previous reply to comments: l. 195-196).

Specific comment From reviewer: l. 336 Our reply: we moved the abbreviation FDrc to the end of the sentence where it directly is linked to Table 4. We feel that the tables 1 and 4 together with the update text make the text readable enough and an additional list of abbreviations is not required.

Specific comment From reviewer: l. 345: Our reply: capital letters were deleted

Specific comment From reviewer: l. 349-350: Our reply: in the text we use "the Netherlands", only in References we allowed "The"

Specific comment From reviewer: l. 372: Our reply: see our reply to Specific comment From reviewer: l. 336

Specific comment From reviewer: l. 383: Our reply: Sentence was extended to clarify the differences : "differences between downward flux across the bottom of the rootzone (qpercolation in Figure 2) of 3 hydrological conditions"

Specific comment From reviewer: l. 393, 394: Our reply: clearly was deleted

Specific comment From reviewer: l. 421-422: Our reply: We started with the statement

"High upward flow values are found in loamy soils as is expected". Low values for loamy soils were found because of high water stress which reduce upward flow and can be more important than soil type. The sentence was slightly rephrased to make this more clear.

Specific comment From reviewer: l. 430: Our reply: We looked at figures 46 and 47 in Tolman (1937, p.157) and indeed mention that we did-not consider hysteresis. Perhaps an additional study might zoom in on this aspect. Reference: Tolman, C. F. (1937). Ground Water. Retrieved from https://ia801500.us.archive.org/16/items/in.ernet.dli.2015.1788/2015.1788.Ground-Water.pdf

Specific comment From reviewer: l. 452: Our reply: "then" changed to "than"

Specific comment From reviewer: l. 460: Our reply: we changed "difference" into "impact of upward flow on crop yields" because it is more clear and that is the core of our message.

Specific comment From reviewer: l. 494, Our reply: all References have year in parenthesis

Specific comment From reviewer: about definitions in Tables 1, 5 and 6 Our reply: Table 1 was already updated; Tables 5 and 6 have now explanation of FDnc, FDrc and Ave

Specific comment From reviewer about Figures 4, 5, 6, and 7:: Our reply: Number on the y-axis are rotated by 900 , so they face the reader now.

Please also note the supplement to this comment:
https://www.hydrol-earth-syst-sci-discuss.net/hess-2017-223/hess-2017-223-AC1-supplement.pdf
* * *
223, 2017.

**Supplement:**

[revised manuscript text omitted]

---

## Referee Comment (RC2) · Anonymous Referee #2 · 30 Jul 2017

In this manuscript, the authors quantify the impact of capillary rise and water vertical recirculation on yield through simulations with the software SWAP. The model is calibrated for maize, grassland and potato over several years and places throughout the Netherlands. In order to isolate the respective impact of capillary rise from the groundwater, and water recirculation, the simulation boundary condition is modified from average fluctuating groundwater ("Ave") to free drainage ("FDrc") to free drainage with no upward water recirculation at the lower limit of the root zone ("FDnc").

General comments:

The paper is well written and the tables and figures are clear. However, substantial effort would be necessary improve the methods and properly discuss the results in light of the existing literature.

Throughout the introduction, the authors make the points (i) that capillary rise and recirculation are different things, (ii) that they both contribute to crop yield, (iii) that a significant part of crop models use a "bucket approach" to soil water storage (with no capillary rise/recirculation), and (iv) that capillary rise and recirculation should be included in crop models as they significantly contribute to yield. I think that the latter point is of great interest (does this extra process significantly improve the predictive power of the crop model?), and demonstrating it for several crops and soils seems to be the core idea of the paper. Here I have a first major concern because such a demonstration would necessitate comparing the accuracy of yields predicted with the bucket approach versus the proposed approach, which was not properly done in this study and would imply quite substantial additions to the manuscript. Among other additions: both approaches would need independent calibrations; the validation stage should involve an independent partition of the observed data (not already used in the calibration stage); proper statistical analyses should be used to determine if the added process and parameters significantly improve the predicted yield.

My second major concern is that the description of the methods is rather incomplete, as central elements of the study are not presented. How is the numerical method blocking recirculation implemented? What is the domain vertical discretization? How is yield affected by water limitation in SWAP (provide at least a couple of sentences)? How were the 72 sets of Mualem – van Genuchten parameters obtained (e.g. artificial neural network, pedotransfer function, etc.)?

In the results, the authors (i) claim that the differences between simulated and observed yields are within acceptable range, and then (ii) use simulated yields to draw more general conclusions on the relative contribution of upward water flow to yield. My third major concern is that the quality of the yield model is not as good as suggested by the mean error indicator, which is the indicator mostly referred to throughout the results and discussion. In the mean error indicator, positive and negative errors cancel each other, so that a model with random predictions but the same average value as the

average observation would have a null mean error. Hence, the mean error is not such a good model quality indicator. In Table 2, 4 out of 5 Nash-Sutcliffe indices for yield are below zero, which suggests that the yield model is not accurate. Furthermore, the yield model root mean square errors (10% to 25%) exceed the yield model sensitivity to upward flow (2% to 22% in Table 3). Given the low certainty on yield predictions, I do not think that the impact of upward flow on yield can be discussed with confidence.

Specific comments:

Title: The title should mention that the paper presents simulated results. I think that "Impact of capillary rise and recirculation on simulated crop yields in SWAP" would be more representative of the current content of the manuscript.

Line 31 (L31): It is unclear what the unit "a" represents. Time units are year and season throughout the paper, but it does not seem to correspond to any of them.

L137: The chosen time step is one day, which seems inappropriate considering that recirculation reportedly varies largely within a day-night cycle [Li et al., 2002; Guderle and Hildebrandt, 2015]. How does the chosen time step affect the quantified recirculation? It should be discussed, and could easily be tested in the model I guess. Yet, at line 269 the authors mention variable time steps lower than a day. Please clarify from the beginning.

L183: Feddes et al. [1978] stress function and Jarvis [1989] compensation function are known to be entangled in such a way that they simultaneously affect plant water stress [Javaux et al., 2013], and thus (I guess) yield in SWAP. However, only Feddes et al. [1978] stress function parameters are presented in Table 1. Please provide compensation parameters too.

L252 and many other places: The term "upward flow" is ambiguous as capillary rise and upward recirculation are both upward flow. Please clarify if one of them, or both, is/are concerned.

L409-L434: Please use references to support your claims.

Typos:

L96: "ss"?

References:

Feddes, R. A., Kowalik, P. J., and Zaradny, H.: Simulation of Field Water Use and Crop Yield, edited by: Pudoc, 189 pp., 1978.

Guderle, M., and Hildebrandt, A. (2015), Using measured soil water contents to estimate evapotranspiration and root water uptake profiles – a comparative study, Hydrol. Earth Syst. Sci., 19, 409-425, 10.5194/hess-19-409-2015.

Jarvis, N. J. (1989), A simple empirical model of root water uptake, J. Hydrol., 107, 57-72.

Javaux, M., Couvreur, V., Vanderborght, J., and Vereecken, H. (2013), Root Water Uptake: From 3D Biophysical Processes to Macroscopic Modeling Approaches, Vadose Zone J., 12, 16 pp., doi:10.2136/vzj2013.02.0042.

Li, Y., Fuchs, M., Cohen, S., Cohen, Y., and Wallach, R. (2002), Water uptake profile response of corn to soil moisture depletion, Plant Cell Environ., 25, 491-500, 10.1046/j.1365-3040.2002.00825.x.

---

## Referee Comment (RC3) · P. Kowalik (Referee) · 9 Aug 2017

**P. Kowalik (Referee)**

piotr\_kowalik@tlen.pl

Received and published: 9 August 2017

Authors are: Joop Kroes, Ivan Supit, Jos Van Dam, Paul Van Walsum, Martin Mulder, all from the Wageningen University, what is the proof of the high quality paper. Title is: "Impact of capillary rise and recirculation on crop yields". I do not agree with the term "recirculation" here, much better would be "water retention". The paper is describing of influence of soil water on grass, maize and potato yields in the Netherlands. The idea is to describe upward capillary rise and retention of percolating water and the crop yield. The main idea was published by Feddes et al (1978) in the book "Simulation of field water use and crop yield". The authors are following the idea from this monograph, but with many new concepts and ideas. The starting point is the Richards equation with the sink term, including the root water uptake. The water is influencing the growth

of crops. In the monograph of Feddes et al. (1978) crop yield is simulated by CROP model according to the concept of Cornelius Theo de Wit. The same concept is repeated in the following model called WOFOST (1986). I would suggest to add some remarks about CROP model which is still in use in many countries (Finland, Sweden, Poland, Italy), and model WOFOST is more complicated. The numerical experiments are important, but "a synthetic modelling option has been implemented to stop upward flow reaching the root zone without inhibiting percolation" and it is not logical and realistic. The fantasy of authors is too great here. The results are proper and the model is well implemented and validated. If the authors are stopping upward flow the yield of crops is greatly reduced, what it well presented. I agree with the conclusions. They write "we think that the quantification of upward flow on yield is a novelty". Most of the conclusions may be accepted. The paper may be published after minor revision. The possible changes are indicated above.

---

## Author Comment (AC2) · 30 Aug 2017

Specific comment From reviewer: Line 430: Our reply: We looked in more detail at figures 46 and 47 in Tolman (1937, p.157) and concluded that this type of macro-scale hysteresis is automatically accounted for by the dynamic numerical scheme of SWAP and does not require any special modelling option for hysteresis of the $\theta$-h relationship. Reference: Tolman, C. F. (1937). Ground Water. Retrieved from https://ia801500.us.archive.org/16/items/in.ernet.dli.2015.1788/2015.1788.Ground-Water.pdf

---

## Author Comment (AC3) · 30 Aug 2017

Reviewer's first major concern: "such a demonstration would necessitate comparing the accuracy of yields predicted with the bucket approach versus the proposed approach, which was not properly done in this study and would imply quite substantial additions to the manuscript."

Our reply to the first major concern: The idea of the reviewer is strongly appealing and we would have done so if the measurements at the experiments were sufficient. However, the measured data sets are insufficient to calibrate and validate the soil and crop parameters in such detail that they allow proper statistical evaluation of the bucket approach and the approach with full simulation of capillary rise and recirculation. The calibration of both model approaches has too much freedom with the available datasets, which upsets a reliable validation. Therefore we used the measured data sets to illustrate that with common soil and crop input values SWAP-WOFOST yields realistic and plausible results for the crops considered in this study. Further, crop growth and soil water flow are simulated by SWAP-WOFOST with state of the art concepts. Therefore we may expect that the model itself can be used to show the effect on crop yield of different boundary conditions with respect to zero flux, recirculation and capillary rise.

Reviewer's second major concern: "description of the methods is rather incomplete, as central elements of the study are not presented" and in detail: "How is the numerical method blocking recirculation implemented? What is the domain vertical discretization? How is yield affected by water limitation in SWAP (provide at least a couple of sentences)? How were the 72 sets of Mualem – van Genuchten parameters obtained (e.g. artificial neural network, pedotransfer function, etc.)?"

Our reply to the second major concern: The numerical method to block recirculation is explained in line 252-256: "A synthetic modelling option has been implemented to stop upward flow from reaching the root zone, without inhibiting percolation. This option is implemented in the numerical solution of the Richards equation and minimizes vertical conductivity just below the root zone in situations that the model simulates upward vertical flow." However, this explanation is not very detailed and we will therefore add some user suggestions in the Supplementary Material that is part of this paper. We implemented this option in SWAP version 4 which is available online since June 2017 (www.swap.alterra.nl). Our explanation in the Supplementary Material will also serve as support to the user manual (Kroes et al., 2017). This should increase transparency and will allow future users to simulate similar conditions.

The domain of the vertical discretization is briefly summarized in line 234-250. We applied 72 different soil schematizations which is explained in line 226-237. Each soil schematization consists of one or more soil horizons, each with different soil physical/hydraulic properties. This is described in detail by Wösten et al. (2013a)

HESSD
and available on internet http://www.wur.nl/nl/show/Bodemfysische-Eenhedenkaart-BOFEK2012.htm. We will add a sentence to further clarify this.

We will insert sentences to clarify how yield is affected by water limitation. In line 200 we will insert: "Oxygen and drought stress cause water limitation which directly influences yields through reduced transpiration. Drought stress in SWAP is described by the dry part of the reduction function proposed by Feddes et al. (1978). Oxygen stress with the process-based method of Bartholomeus et al. (2008)." This reference will be added to the reference list: Bartholomeus, R. P., Witte, J. P. M., van Bodegom, P. M., van Dam, J. C., and Aerts, R., (2008). Critical soil conditions for oxygen stress to plant roots: substituting the Feddes-function by a process-based model, J. Hydrol., 360, 147–165

Reviewer's third major concern: "the quality of the yield model is not as good as suggested by the mean error indicator, which is the indicator mostly referred to throughout the results and discussion." And "the mean error is not such a good model quality indicator. In Table 2, 4 out of 5 Nash-Sutcliffe indices for yield are below zero, which suggests that the yield model is not accurate" A negative NS index means that a the observed mean is a better predictor than the model outcome.

Our reply to the third major concern: The reviewer has a point. The simulation results could be improved. The negative NS index demonstrates that the observed mean is a better predictor than the model outcome. However, one has to bear in mind that perfect calibration is not the objective of this study. As mentioned in line 198, we used calibration values from earlier studies (Kroes et al. 2015 and Hack et al., 2016). No detailed assimilation measurements were executed on the fields and the meteorological data was not measured on site, but taken from meteorological stations sometimes more than 30km away. Furthermore, no detailed information concerning fertilizer applications and soil carbon is available, therefore we considered it constant in time. In lines 311-314 we mention that the calibration can be improved.
In this research we want to demonstrate that for crop growth modelling a proper soil modelling component that includes recirculation is important. We know from experience that WOFOST with a simple soil model underestimates water availability in the rooting zone and consequently overestimates drought stress (A regional implementation of WOFOST for calculating yield gaps of autumn-sown wheat across the European Union. Van Boogaard, H., Wolf, J., Supit, I., Niemeyer, S., Ittersum, M., 2013. Field Crops Research, vol143, March 2013, p. 130-142). The results in this study clearly demonstrates that the (simulated) yield reduction resulting from over-estimated drought stress is reduced when capillary rise and recirculation are taken into account. We will add a reference to the above mentioned paper to the reference list.

Specific comment From reviewer

Title: The title should mention that the paper presents simulated results. Our reply: "Impact of capillary rise and recirculation on simulated crop yields" is fine

Line 31 (L31): It is unclear what the unit "a" represents Our reply: the unit "a" is changed into the unit "year" or the unit "season"

Line 137: The chosen time step is one day Our reply: For crop growth WOFOST applies a time step of one day with integration of light interception during the day. However, SWAP calculates soil water flow at very small time steps ranging between a few seconds and a few hours depending on the variation in boundary conditions. The exchange of information between the 2 sub models SWAP and WOFOST occurs at the end of day and therefore does not allow impact analyses during the day. Such detailed analyses would require a different crop growth model.

Line 183: Feddes et al. (1978) stress function and Jarvis (1989) compensation function are known to be entangled Our reply: For drought stress we applied Feddes et al. (1978) combined with Jarvis (1989) compensation function using a compensation factor (ALPHACRIT) of 0.7 (see manual Kroes et al., 2017). The compensation factor will be added to the text. HESSD
Line 252 and many other places: The term "upward flow" is ambiguous as capillary rise and upward recirculation are both upward flow Our reply: you are very right that "upward flow" may have 2 meanings. It may refer both to recirculation and capillary rise. Previous reviewers indicated this and made us introduce Figure 2. We will screen the text on "upward flow" and use EITHER recirculation OR capillary rise. Capillary rise always includes recirculation. We will only speak of "upward flow" when it can refer to both "recirculation" and "capillary rise".

Line 409-434: Please use references to support your claims. Our reply: we will use references to the Tables with simulated results.

Typos: Line 96: "ss"? Our reply: "ss" in Line 96 will be changed into "as"

References: Our reply: Thank you for the suggestions. These references are relevant and we will give them a proper place in the manuscript.

---

## Author Response (AR3)

**Response to anonymous Referee # 2 (Report # 1,  submitted on 16 Dec 2017)**

*Referee's main concern*: "I went through the comments of the referees and replies of the authors. While minor comments were appropriately dealt with, most of the major points raised by the referees were incompletely addressed."

*Our reply to the main concern:*
We read the main concern and agree with many points. We gave answers to the concern points raised by Referee #2 in our replies below:

*Referee's first concern:*
"As reported by referees #1 in his/her general comments, "the authors should recognize that quantitative work was done on the relation between capillary rise and crop yield". To this remark, I would add that the authors should extensively discuss their results in comparison to the existing literature. The referee #1 specifically referred to such studies and pointed at misleading sentences suggesting that there were no such studies. Yet, these misleading sentences are mostly left unchanged in the manuscript, for instance in the conclusions "We think that the quantification of upward flow on yield is a novelty" and "Another aspect which cannot be found in the referenced studies is the lack of a quantification of the impact of capillary rise and recirculation on crop yields".

*Our reply to the first concern:*
Referee #1 found our sentences about "quantification of capillary rise and recirculation" not correct because we did not refer enough to existing literature, especially to the early Dutch physical scientist, Symen Barend Hooghoudt. He had a good point and we corrected that. We re-read Hooghoudt (1937) writings about capillary rise, groundwater and potential theory and found no references to recirculation and also no reference linked to a quantification of yields. Other relevant literature on the relation between capillary rise and crop yield is described in lines 49-96 of the Introduction. As our literature search is still limited, we want to avoid "misleading sentences", so we changed the words "quantification of upward flow" into "quantification of recirculation" in lines 513-515 and line 519 to be more specific and avoid confusion.

*Referee's second concern*:
"The second general comment of referee #1 points out that the illustrations of hydrological conditions in Figure 2 are unclear, particularly the first two conditions. Yet, the figure was left unchanged. Please clarify the figure, and note that the red horizontal bars can be easily removed by setting the Powerpoint to full screen before pushing on the PrintScreen button."

*Our reply to the second concern:*
As authors we discussed Figure 2 various times, also in response to the reviewers comments, and current Figure 2 in the last manuscript is the outcome of these discussions. In the caption and text we explained the first two conditions as clearly as possible: 'Conditions a and b have free-draining bottom

boundary conditions without groundwater. Condition a is artificially created to explicitly demonstrate the role of recirculating percolation resulting in upward flow to the root zone. Condition b is a common free drainage situation which includes upward flow due to recirculating percolation water.'

Referee #1 states "I do not understand Figure 2b. The authors show no impervious layer at the bottom of the figure. So how can water move upward by "recirculation"? Without an impervious layer, it seems to me that Figure 2b should be the same as Figure 2a." Both conditions a and b have free drainage at the bottom of the soil profile. Also without impervious layer, water may move upward below the root zone due to hydraulic head gradients which causes recirculation.

We maintain the definitions of the fluxes in the Figure 2 ($q_{percolation}$, $q_{recirc}$, and $q_{caprise}$) as these definitions and symbols are used throughout the Tables and discussion of the results. In reaction to the reviewers comment we already added to the text that an implicit scheme is used for the hydraulic conductivity to implement the synthetic modeling option. In the new version we will add that the free drainage option is applied at a depth of 5.5 m, in order to address the reviewers concern with respect to the effect of the length of the soil column on the recirculation flux.

It is not clear to us why red horizontal bars are mentioned at Figure 2: we don't see these bars on our screen.

*Referee's third concern*:
"The first major comment of referee #2 highlights that the methods presented in the study are not adequate to reach its objective (i.e. demonstrating that, in place of the common "bucket approach", recirculation should be implemented in crop models in order for them to be more accurate). A comparison of simulated results obtained with both assumptions is proposed. The comparison does not demonstrate that recirculation makes the crop model more accurate, but only that Swap is sensitive to recirculation. In their reply, the authors explain that the data available is not sufficient to validate that Swap with recirculation is more accurate than "bucket" Swap, and stick with the sensitivity analysis. Their hypothesis thus cannot be demonstrated."

*Our reply to the third concern:*
Our hypothesis is indeed that the process of recirculation makes crop modelling more accurate. To demonstrate and support our hypothesis we added another case study. This is reported in section S5 of the supplementary material. In this section we demonstrate the difference in soil water pressure head in the upper part of the root zone as caused by drying of the soil due to a lack of recirculating water in the hydrological condition (main text fig 2a). This results in a lowering of average yields with 609 kg/ha (from 7132 to 7741 kg/ha DM, which is about 9% yield reduction due to recirculation. To our opinion this supports the hypothesis that it is recommended to use tools that support this process of recirculation in conditions where the vertical water fluxes across the root zone is relatively high. This will clearly be the case in delta regions where periodically you have a precipitation excess. We extended the Discussion with these remarks.

*Referee's fourth concern*:
"The third major concern of referee #2 addresses the validity of lessons drawn from the simulated results, given that the accuracy of the model with recirculation, compared to measured yield, is so low.

Switching recirculation on or off generates deviations of the predicted yield that are all smaller than the standard error of the model. The authors reply that the poor quality of the input data might explain the poor accuracy of the model, and insist on the point that they know *by experience* that recirculation is important in crop models. They might be right but the problem is that the results shown in the manuscript do not support the validity of the model *in the considered conditions*, and support even less so the validity of lesson taken from the model in conditions where it appears not to be accurate. It seems that the authors take predictions of the model for granted, as if they were real observations."

*Our reply to the fourth concern:*

The crop growth experiments which we simulated did lack various input parameters with respect to soil, water and plant conditions. We decided not to calibrate the model to the measured soil water and crop growth data. Calibration would not be a big deal given the large number of input parameters that are used. However, this would make the model tuned to very specific soil/year combinations, for which the accuracy of the soil water and crop yield observations was not clear. Rather, we decided to use general soil and plant parameters, and showed for the particular fields and years the simulated soil water conditions and crop yields are realistic.

Recirculation itself is calculated with the Richards' equation, which is considered among soil physicists the reference equation for soil water movement. To analyse the effect of recirculation, we implemented the extra simulation option in which only percolation fluxes are allowed, but upward soil water fluxes are prohibited. In an additional case study (Supplementary Material Section 5) we showed with observations that switching off recirculation results in less accurate pressure head and crop yield predictions.

In paragraph 3.2 we performed simulations for 45 years and 72 soils, so we may assume that the simulated effect of switching on/off recirculation are representative for climate and soil conditions in The Netherlands.

To quantify the effect of redistribution in Tables 4-6, we indeed consider the model simulations with redistribution as real observations. In this point we agree with the reviewer. We state this in the paper with the sentence in line 452: "we may expect that the model itself can be used to show the effect on crop yield of different boundary conditions with respect to zero flux, recirculation and capillary rise."

*Referee's fifth concern*:

"Referee #3 points out that allowing percolation while blocking upward flow is not a realistic condition. I agree with the reply of the authors, that the point is not to make the boundary condition realistic, but to make the modified model representative of a bucket model. However, the quality of two models cannot be compared by using the parametrization of only of them. Bucket models may be quite effective in reproducing observed yields, despite their lack of physical basis, when using their own parametrization. In this paper the authors try to discredit the bucket approach in a way that is not scientifically sound."

*Our reply to the fifth concern:*

It is not our intention to discredit bucket approaches. To makes this clear we applied the following changes:
- In line 101, 266 and 437 we change "simple ''bucket" into "''bucket"
- We eliminated in line 466 "Furthermore we know from experience that WOFOST"

- We introduced after line 468 "We suggest to generate additional relations about the contribution of recirculation and capillary rise to upward flow to the root zone. Such an approach has been used in AQUACROP to derive a relation between capillary rise and groundwater (Van Gaelen et al., 2017). Another approach is to calibrate the conceptual parameters of a bucket model with agro-hydrological models like SWAP as done by Romano et al. (2011)."
- In line 474 we eliminated the line "An adequate soil schematization is relevant for all models but especially for those that use a bucket approach."

**Response to anonymous Referee # 4 (Report # 2, submitted on 26 Jan 2018)**

5   *Referee's main concern*: "Now the main concerns are the Abstract, quality of some of the Figures, and some verb tense issues.."

   *Our reply to the main concern:*
   We followed practically all suggestions given by the Referee in our replies below

   *Referee's concern about abstract*:
   "Abstract: Start with a justification statement (why the research is needed). Then include a hypothesis,
15  objective, or proposed
   outcome statement (separate from methods statements).
   "Upward soil water flow is a vital supply of water to crops. The purpose of this study was to determine if upward flow and
   recirculated percolation water can be quantified separately, and to determine the contribution of
20  capillary rise and recirculated
   water to crop yield and groundwater recharge." Other parts of the abstract may then need to be shortened."

   *Our reply to the concern about abstract:*
25  We thank the referee and used his proposed sentence for the abstract.

   *Referee's concern about Method*:
30  "Present your methods in past tense"

   *Our reply to the concern about Method:*
   We agree and adjusted it throughout the text. We used past tense when activities are finished and present tense for statement which are still valid.

   *Referee's concern about Results,* :
   "Figure 4 : Either remove the caption on the lower plot (since it is the same as for the upper plot) or
40  enlarge it."
   "Present your results in past tense."

   *Our reply to the concern about Results:*
   We removed the legend on the lower plot
45  We used past tense when activities are finished and present tense for statement which are still valid.

*Referee's concern about Supplemental Material* :
"Supplemental Figures S4-S18 needs axes, tick labels, data points, plot captions, etc. to be greatly enlarged because they are
undecipherable."

*Our reply to the concern about Supplemental Material:*
We adjusted Figures S4-S18 in the Supplementary Material

*Referee's concern about L. 377-379:*
"L. 377-379: This sentence is awkward, perhaps "The shallow groundwater in Dutch conditions often do not have deep leaching because excess precipitation or upward seepage is discharged via drainage systems." (Possibly add a citation.)"

*Our reply to the concern about L. 377-379:*
We agree that the sentence in L377-379 is not clear and changed the sentence using the proposal of the referee.

*Referee's concern about Discussion and Conclusions:*
"Present your methods in past tense"

*Our reply to the concern about Discussion and Conclusions:*
We used past tense when activities are finished and present tense for statement which are still valid.

*Title:*

**Impact of capillary rise and recirculation on simulated crop yields**

*Joop Kroes[1], Iwan Supit[1,2] Jos Van Dam[3], Paul Van Walsum[1], Martin Mulder[1]*

**[1]** *Wageningen University & Research - Environmental Research (Alterra)*

**[2]** *Wageningen University & Research – Chair Water Systems and Global Change*

*[3] Wageningen University & Research – Chair Soil Physics and Land Management*

*Abstract*

Upward soil water flow is a vital supply of water to crops. The purpose of this study was to determine if upward flow and recirculated percolation water can be quantified separately, and to determine 
[revised manuscript text omitted]
.  The shallow groundwater in Dutch conditions (Figure 2 c)  often does not have leaching at greater depth  because excess  precipitation or upward seepage is discharged via drainage systems. The average condition we used had no leaching but seepage of 227, 155 and 291 mm.year$^{-1}$ for grassland, maize and potatoes (Table 4, $q_{seepage}$).

As can be expected, the synthetic condition without upward flow and without groundwater (Figure 2 a), had the lowest simulated mean yields for all crops (Table 4). The highest mean yields were simulated when average groundwater situations including capillary rise were considered (Table 4, Ave). The relative mean yield increase was lowest for maize and highest for grassland (Table 5) which was probably caused by the difference in rooting depth.

[revised manuscript text omitted]

465     simulated by SWAP-WOFOST with state of the art concepts. Therefore we may expect that the model itself can be used to show the effect on crop yield of different boundary conditions with respect to zero flux, recirculation and capillary rise.
Our hypothesis is that the process of recirculation makes crop modelling more accurate.
To demonstrate and support our hypothesis we added another case study. This is reported in

470     section S5 of the supplementary material. In this section we demonstrated the difference in soil water pressure head in the upper part of the root zone as caused by drying of the soil due to a lack of recirculating water in the hydrological condition (Figure 2a). This resulted in a lowering of average yields with 609 kg/ha (from 7132 to 7741 kg/ha DM, which is about 9% yield reduction due to recirculation. This supports the recommendation to use tools that

475     support this process of recirculation in conditions where the vertical water fluxes across the root zone is relatively high. This will clearly be the case in delta regions where you have occasionally a precipitation excess.

Furthermore we know from experience that WOFOST with aA bucket approach generally

480     underestimates water availability in the rooting zone and consequently overestimates drought stress (Boogaard et al., 2013). We suggest to generate additional relations about the contribution of recirculation and capillary rise to upward flow to the root zone. Such an approach has been used in AQUACROP to derive a relation between capillary rise and groundwater (Van Gaelen et al., 2017). Another approach is to calibrate the conceptual

485     parameters of a bucket model with agro-hydrological models like SWAP as done by Romano et al. (2011).

Our analysis shows that soil properties and soil profile layering are important because differences in soil hydraulic properties influence vertical water flow. High upward flow values were found in loamy soils as was expected (Table 6, max row), but if water stress was high and upward flow was low the influence of soil type decrease and low upward flow values were found for loamy soils (Table 6, min row).  Comparing the minimum yield values it shows that there was a large difference between these soil types in free-drainage conditions with and without upward flow. This means that the storage capacity of loamy soils was larger than the one of sandy soils as could be expected. The yield variation between soil types in water stress conditions was large and illustrate the need for a proper soil schematization especially in stress full hydrological conditions.  As the influence of recirculation increase, the yield variation becomes less and the influence of soil type decrease. In situations without water stress the soil type was less important. In conditions where groundwater and capillary rise occurs (Ave) yield variation was hardly influenced by soil type.

Therefore Modelling concepts should consider dynamic interactions between soil water and crop growth. Crop models in general should consider recirculation of soil water and, especially in low lying regions like deltas, groundwater dynamics should be considered as well.

Precipitation, soil texture and water table depth jointly affected the amount of groundwater recharge and time-lag between water input and groundwater recharge (Ma et al., 2015). We quantified some of these issues, but several items remain, such as the impact of rooting depth on crop yield and transpiration. Also soil and water management practises like ploughing and irrigation, were not considered. Furthermore the rooting pattern needs a more detailed analysis; we applied an exponential decrease of root density and compensation of root uptake according to Jarvis (2011) but the macroscopic root water uptake concept was still simple and may require a more detailed analyses (Dos Santos et al. 2017). Another item we neglected is the preferential flow of water by the occurrence of non-capillary sized macropores (Bouma, 1961, Feddes, 1988), which is relevant in especially clay soils. Hysteresis of the water retention function was also not considered. An additional analysis of these issues is recommended, especially the impact of different rooting patterns on capillary rise should be addressed.

The impact of soil type on yield increase when environmental conditions became dryer; situations without groundwater and without recirculation had less yield and higher yield variation than situations where groundwater influenced capillary rise (For detailed information on results see the supplementary material S1 and S3).

*5. Conclusions*

530 We quantified the impact of upward flow on crop yields of grassland, maize and potatoes in layered soils. We compared situations with average groundwater levels with free-drainage conditions with and without upward flow. The largest impact of upward flow on crop yields was found when one compares situations with average groundwater levels with free drainage conditions without upward flow. From these differences one may conclude that neglecting

535 upward flow has a large impact on simulated yields and water balance calculations especially in regions where shallow groundwater occurs. The comparison showed long term average yield-reductions of grassland, maize and potatoes of respectively 26, 3 and 14 % (Table 5) or respectively 3.7, 0.3 and 1.5 ton Dry Matter per ha (Table 4). Reduction of the percolation flux can be considerable; for grassland and potatoes the reduction was 17 and 46% (Table 5) or

540 63 and 34 mm (Table 4).

About half of the yield increases was caused by internal recirculation as occurs in free-drainage conditions and the other half was caused by an increased upward capillary flow from groundwater. Improved modelling should consider upward flow of soil water which will result in improved estimates of crop yield and percolation.

545 We think that the quantification of recirculation  on yield is a novelty, especially  recirculation as part of upward flow and its relation to 
[revised manuscript text omitted]

Romano, N., Palladino, M., & Chirico, G. B. (2011). Parameterization of a bucket model for soil-vegetation-atmosphere modeling under seasonal climatic regimes. Hydrology and Earth System Sciences, 15(12), 3877–3893. http://doi.org/10.5194/hess-15-3877-2011

685

690

695

700

705

710

715

Rötter, R. (1993). Simulation of the biophysical limitations to maize production under rainfed conditions in Kenya. Evaluation and application of the Model WOFOST. Materiel zur
720        Ost-Afrika vorschung, Heft 12, pp. 261 + Annexes.

Šimůnek, J., Sejna, M., Saito, H., Sakai, M., & Genuchten, M. T. van. (2008). The HYDRUS-1D Software Package for Simulating the One-Dimensional Movement of Water, Heat, and Multiple Solutes in Variably-Saturated Media, Version 4.0 April 2008. Environmental Sciences. RIVERSIDE, CALIFORNIA.

725    SSSA (2008). Glossary of Soil Science Terms 2008. Soil Science Society of America. Soil Science. Retrieved from https://www.soils.org/publications/soils-glossary#

Supit, I. (2000). An exploratory study to improve predictive capacity of the Crop Growth Monitoring System as applied by the European Commission. Treebook 4. Treemail publishers. Retrieved from www.treemail.nl

730    Supit, I., van Diepen, C. A., de Wit, A. J. W., Wolf, J., Kabat, P., Baruth, B., & Ludwig, F. (2012). Assessing climate change effects on European crop yields using the Crop Growth Monitoring System and a weather generator. Agricultural and Forest Meteorology, 164, 96–111.

Talebnejad, R., & Sepaskhah, A. R. (2015). Effect of different saline groundwater depths
735        and irrigation water salinities on yield and water use of quinoa in lysimeter. Agricultural Water Management, 148, 177–188. http://doi.org/10.1016/j.agwat.2014.10.005

Van Bakel, P.J.T., Massop, H.Th.L., Kroes, J.G., Hoogewoud, J., Pastoors, R., Kroon, T., (2008). Actualisatie Hydrologie voor STONE 2.3; Aanpassing randvoorwaarden en parameters, koppeling tussen NAGROM en SWAP, en plausibiliteitstoets. (Eng:
740        Updating the hydrology component in STONE 2.3; Adjusting boundary conditions and parameters, linking NAGROM and SWAP, and plausibility test). WOt-rapport 57. Wettelijke Onderzoekstaken Natuur & Milieu (MNP), Alterra, Wageningen, The Netherlands.

Van Dam, J. C., Groenendijk, P., Hendriks, R. F. A., & Kroes, J. G. (2008). Advances of
745        Modeling Water Flow in Variably Saturated Soils with SWAP. Vadose Zone Journal, 7(2), 640–653. http://doi.org/10.2136/vzj2007.0060

Van Diepen, C.A. van, Wolf, J., Keulen, H. van (1989). WOFOST: a simulation model of crop production. Soil Use and Management, 5:16-24.

Van den Brande, M. (2013). Remote sensing beelden van NDVI en hydrologisch
750        modelleren. BSc thesis Wageningen UR. Wageningen.

Van der Gaast, J. W., Massop, H. T. L., & Vroon, H. R. J. (2009). Effecten van klimaatverandering op de watervraag in de Nederlandse groene ruimte. Alterra-rapport 1791. Wageningen. Retrieved from www.alterra.nl

Van der Ploeg, M. J., & Teuling, A. J. (2013). Going Back to the Roots: The Need to Link Plant Functional Biology with Vadose Zone Processes. Procedia Environmental Sciences, 19, 379–383. http://doi.org/10.1016/j.proenv.2013.06.043.

Van Gaelen, H., Vanuytrecht, E., Willems, P., Diels, J., & Raes, D. (2017). Bridging rigorous assessment of water availability from field to catchment scale with a parsimonious agro-hydrological model. Environmental Modelling and Software, 94, 140–156. http://doi.org/10.1016/j.envsoft.2017.02.014

[revised manuscript text omitted]